# Robust immune cell infiltration and macrophage senescence occur within a week of recovery after limb immobilization in older adult skeletal muscle

Chad M. Skiles[1], Zachary J. Fennel[1], Paul-Emile Bourrant[2], Elena M. Yee[2], Robert J. Castro[2], Hannah A. Zabriskie[1], Melina Itinose[1], Mark A. Supiano[4], Ryan M. O'Connell[5] and Micah J. Drummond[1,2,3]

[1]*Department of Physical Therapy and Athletic Training, University of Utah, Salt Lake City, UT, USA*
[2]*Department of Nutrition and Integrative Physiology, University of Utah, Salt Lake City, UT, USA*
[3]*Molecular Medicine Program, University of Utah, Salt Lake City, UT, USA*
[4]*Department of Internal Medicine, Division of Geriatrics, Center on Aging, University of Utah, Salt Lake City, UT, USA*
[5]*Department of Pathology, Division of Microbiology and Immunology, University of Utah, Salt Lake City, UT, USA*

Handling Editors: Karyn Hamilton & Nima Gharahdaghi

The peer review history is available in the Supporting Information section of this article (https://doi.org/10.1113/JP290346#support-information-section).

*The Journal of Physiology*

**Abstract figure legend** During the 7 days of muscle recovery following 14 days of unilateral limb immobilization, older adults (compared to young) exhibited a robust immune cell infiltration, characterized largely by CD68[+] CD206[+] macrophages. This response coincided with an expansion of senescent macrophages and was associated with excessive collagen deposition. We surmise this altered immune cell response and increased senescent burden impair macrophage function during recovery in aged muscle.

**Abstract**   Immune cells are critical for modulating inflammation and extracellular matrix remodelling for effective muscle regrowth following disuse atrophy. However, disrupted macrophage function and accumulation of cellular senescence may limit muscle recovery in ageing. The present study aimed to compare changes in the cellular dynamics of muscle macrophages, cellular senescence and collagen content during early recovery following 14 days of unilateral limb immobilization in young ($n = 18$; ~24 years) and older male and female adults ($n = 18$; ~69 years). Participants underwent 14 days of immobilization followed by 7 days of re-ambulation. Muscle biopsies were collected at baseline, post-immobilization, and at 2 and 7 days of recovery. During re-ambulation, older adults exhibited elevated immune cell infiltration (haematoxylin and eosin, CD45[+]), higher CD68[+] CD206[+] macrophage content and greater muscle collagen deposition (Sirius Red) compared to their young counterpart. Furthermore, cellular senescence (SA-$\beta$-galactosidase[+]) was elevated, including a high number of macrophages co-labelled with p21 in older skeletal muscle during recovery. At 7 days of recovery, the amount of macrophage infiltration was positively associated with cellular senescence, whereas the senescent macrophage cell population was significantly correlated to the Sirius Red percent area. Our findings suggest that an age-altered immune cell response and the accumulation of senescent macrophages may disrupt collagen remodelling during early muscle recovery following disuse.

(Received 16 October 2025; accepted after revision 25 February 2026; first published online 26 March 2026)

**Corresponding author** M. J. Drummond: Department of Physical Therapy and Athletic Training, University of Utah, Salt Lake City, UT 84112, USA.   Email: micah.drummond@hsc.utah.edu

## Key points

- Age-related impairments in muscle recovery following disuse remain a significant challenge.
- Immune cells, particularly macrophages, play critical role in muscle remodelling.
- Muscle macrophage characteristics during the early phase of recovery following disuse in older adults remain unclear.
- We compared changes in immune cell content, cellular dynamics, cellular senescence, and collagen content during recovery (2 days and 7 days) following 14 days unilateral limb immobilization in young (18–35 years) and older male and female adults (≥60 years).
- During muscle recovery, older adults (*vs.* young), increased muscle collagen content concurrent with an infiltration of macrophages. This response was characterized by a distinct predominance of CD68[+] CD206[+] macrophages and a parallel increase in senescent macrophages.
- Our findings suggest that an altered immune cell response and accumulation of senescent macrophages may disrupt tissue remodelling during muscle recovery in aged muscle.

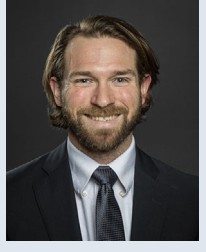

**Chad Skiles** received his Bachelor's Degree in Kinesiology from Lewis-Clark State College. He went on to study Movement and Leisure Sciences under the mentorship of Dr Ann Brown in the Human Performance Laboratory at the University of Idaho, where he completed his master's training. Thereafter, Chad moved to Muncie, Indiana, to pursue doctoral training in human bioenergetics with Dr Scott Trappe at Ball State University. He is currently a post-doctoral fellowship with Dr Micah Drummond, where he investigates the intercellular signalling networks governing myogenic and non-myogenic cell behaviour in ageing skeletal muscle. His work focuses on identifying and testing therapeutic strategies to improve cellular communication and muscle recovery in aged muscle.

## Introduction

Skeletal muscle disuse poses a significant challenge to older adults (OAs), accelerating sarcopenia and its associated detrimental outcomes, including fall risk and the development of chronic diseases (Kehler et al., 2019). Compared to young, OAs typically exhibit slower and often incomplete recovery of skeletal muscle quality and function following periods of disuse (Suetta et al., 2009; Hvid et al., 2010). The mechanisms underlying impaired muscle recovery following disuse atrophy in ageing, especially in humans, are understudied and poorly understood.

Muscle recovery from injury or disuse is a complex process involving sequential inflammatory phases necessary for proper regrowth. Macrophages are early responders to the site of insult, exhibiting a spectrum of inflammatory phenotypes that facilitate debris clearance, extracellular matrix (ECM) remodelling and the regulation of other interstitial cells (e.g., satellite cells, fibro-adipogenic progenitors (FAPs)) essential for muscle remodelling (Chazaud, 2014, 2020; Ahmadi et al., 2022; Fennel et al., 2025). Prior studies in aged rodents demonstrate compromised macrophage-mediated inflammatory profiles and phagocytosis during disuse (White et al., 2015; Reidy et al., 2019b; Fix et al., 2021) and subsequent muscle recovery from injury (Blau et al., 2015; Stahl & Brown, 2015; Tobin et al., 2021). Muscle macrophage dysregulation has similarly been observed in OAs following exercise-induced damage (Mackey et al., 2008; Saclier et al., 2013; Ahmadi et al., 2022; Pfeifer et al., 2024). Macrophage disruption, including reducing satellite cell and FAP activity, hinders proper ECM remodelling and muscle recovery (Lukjanenko et al., 2019; Wang et al., 2019; Uezumi et al., 2021). Although various interstitial cell types are disrupted because of ageing following injury or disuse (Mitchell & Pavlath, 2004; Reidy et al., 2019a; Sorokina et al., 2024; Luo et al., 2025), these findings have been largely limited to pre-clinical models. Muscle cellular dynamics during muscle recovery following disuse in OAs, particularly during the early phase of recovery, remain elusive.

Beyond immune cell dysfunction, ageing muscle often exhibits an accumulation of senescent cells (Zhang et al., 2022). In response to a tissue stressor, cells become senescent as a result of DNA damage or oxidative stress and secrete pro-inflammatory and pro-fibrotic mediators known as the senescence-associated secretory phenotypes (SASP) (Schafer et al., 2020; Zhang et al., 2022; Englund et al., 2023). Although immune cells normally efficiently clear senescent cells to prevent pathological accumulation (Sarig et al., 2019; Da Silva-Alvarez et al., 2020; Young et al., 2022), a decline in immune function with ageing contributes to senescent cell accumulation and exacerbates SASP secretion, ultimately, hindering muscle maintenance and recovery (da Silva et al., 2019; Yousefzadeh et al., 2021; Dungan et al., 2022; Moiseeva et al., 2023; Nolt et al., 2024).

Therefore, the present study aimed to compare the dynamics of myogenic (satellite cells) and non-myogenic cells (immune cells, FAPs), cellular senescence and muscle collagen content during early recovery (2 and 7 days) following unilateral limb immobilization in male and female young and OAs. We hypothesized that there would be age-related differences in cellular dynamics including macrophages, FAPs, satellite cells and senescent cells, that would be related to changes in collagen content during recovery following disuse atrophy.

## Methods

### Subject characteristics and exclusion criteria

Healthy, young adults [YAs: $n = 18$ (10 males/8 females); 18–35 years] and older adults [OAs: $n = 18$ (9 males /9 females); 61–79 years] were recruited from the Salt Lake City area. Characteristics are depicted in **Table 1**. Participants were excluded based on a comprehensive medical history and screening, which included cardiac, pulmonary, hepatic, vascular, haematologic, oncologic and neurologic diseases. Exclusion criteria also encompassed diabetes (HbA1c >6.5%), use of other glucose-lowering and weight loss therapies, chronic kidney disease (serum creatinine >1.5 mg dL$^{-1}$), and uncontrolled hypertension. Screening and experimental procedures were performed at the Clinical and Translational Science Institute (CTSI). This study was approved by the University of Utah Institutional Review Board (IRB #130232) in accordance with the *Declaration of Helsinki*, and all participants provided written informed consent prior to any study procedures. The clinicalTrials.gov ID is NCT04416191.

### Trial design

This study employed a 14 day unilateral limb immobilization protocol followed by a 2 and 7 day recovery period (Fig. 1). At baseline (PRE), lean muscle mass was measured via dual-energy X-ray absorptiometry (Hologic Inc., Discovery Series, Bedford, MA, USA) and a vastus lateralis muscle biopsy was performed on the left leg (designated for immobilization). The left leg was chosen to be the immobilized limb to maximize participation and brace adherence because of personal driving needs. Immobilization was achieved by fitting a knee brace (Orthomen postop knee brace; Orthomen Inc., Foothill Ranch, CA, USA) on the left leg, fixed at 60° flexion to prevent weight bearing activity. Participants were instructed to wear the brace continuously for 14 days,

**Table 1. Subject characteristics**

|  | Young *n* = 18 (10 male/8 female) | Old *n* = 18 (9 male/9 female) |
|---|---|---|
| Age (years) | 24 ± 4 | 69 ± 6 |
| Height (cm) | 175 ± 8 | 171 ± 9 |
| Weight (kg) | 77 ± 12 | 75 ± 12 |
| BMI (kg m$^{-2}$) | 25 ± 2 | 26 ± 3 |
| Glucose (mg dL$^{-1}$) | 90 ± 8 | 94 ± 8 |
| HbA1c (%) | 5.3 ± 0.3 | 5.5 ± 0.3 |

*Note*: Data presented as the mean ± SD.

removing only for personal hygiene (i.e. showering) and sleeping. Crutches or a walker were provided to reinforce non-weight-bearing on the immobilized limb during periods of ambulation. Immediately following the 14 day immobilization, the brace was removed and another vastus lateralis muscle biopsy was performed, followed by lean muscle mass measurements. Subjects were then instructed to return to normal walking behaviours for 7 days yet to avoid any structured exercise program (aerobic/resistance exercise training) during this time. During the recovery phase, additional muscle biopsies were collected on days 2 and 7 of re-ambulation. A final lean muscle mass assessment and a blood draw were obtained on day 7 of recovery. Physical activity via a wrist pedometer (Inspire2; Fitbit, Boston, MA, USA) was determined by assessing average number of steps

7–10 days before the PRE visit, during immobilization (14 days) and during the recovery period (7 days).

### Muscle biopsies

Following a standardized dinner and overnight fast (∼10 h), percutaneous muscle biopsies of the vastus lateralis were performed on the immobilized leg at PRE and at post-immobilization, as well as at the 2 and 7-day-recovery time points. As performed previously, biopsies were collected using a 5 mm Bergstrom needle with manual suction (Reidy et al., 2018; Petrocelli et al., 2023). The biopsy site was marked 10–15 cm proximal to the patella, sterilized with iodine and local anaesthetic (1% lidocaine) was administered. Subsequent biopsies were

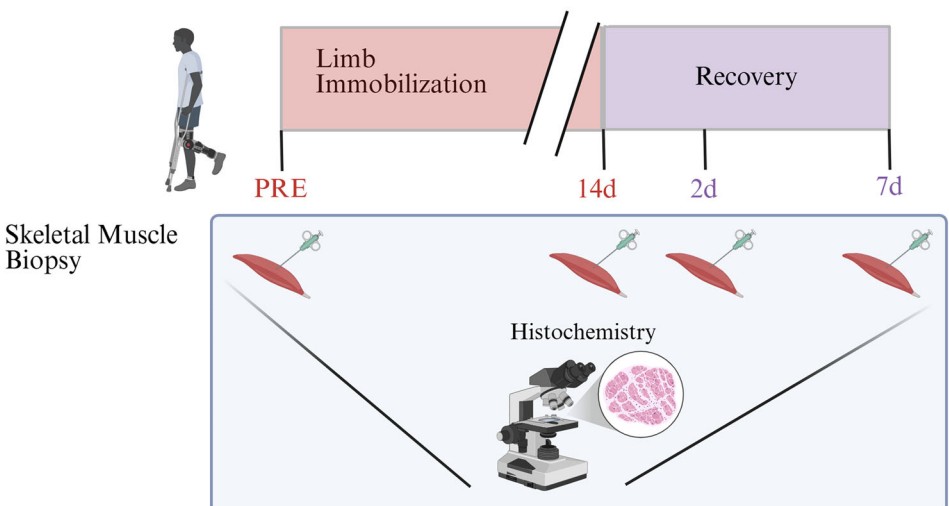

**Figure 1. Trial schematic**
Young and older adults underwent 14 days of immobilization followed by 7 days of recovery. Muscle biopsies were performed on the left vastus lateralis at PRE (pre-immobilization), post-immobilization, and at days 2 and 7 of recovery. During immobilization, participants wore a knee brace on the left leg, which was locked at 60° flexion, and assistive devices (crutches, walker) to ensure non-weight-bearing. After immobilization the knee brace was removed and participants began re-ambulation.

performed ∼3 cm from prior site. Muscle tissue samples (30–50 mg) were carefully oriented to ensure proper fibre directionality, embedded in optimal cutting temperature (i.e. OCT) compound on an aluminium-covered cork, and then frozen in liquid nitrogen-cooled isopentane for 30 s, transferred to dry ice and then stored at −80°C for later analysis.

### Histology and immunofluorescence

Muscle samples were sectioned at 10 μm using a cryostat (Leica Biosystems, Nussloch, Germany) at −25°C and stored at −80°C. Briefly, muscle sections were stained for myosin heavy chain I (catalog. no. BA-D5; DHSB, Iowa City, IA, USA) and IIA (catalog. no. SC-71; DHSB) fibres; satellite cells (Pax7$^+$; catalog. no. AB_528428; DHSB); collagen content (Sirius Red; catalog. no. ab150681; Abcam, Cambridge, UK); immune cell content [haematoxylin (catalog. no. 517-28-2; Sigma-Aldrich, St Louis, MO, USA) and eosin (catalog. no. 17372-87-1; Sigma-Aldrich); CD45$^+$ (catalog. no. ab528428; Abcam)]; FAPs (PDGFR$\alpha^+$; catalog. no. AF-307-NA; R&D Systems, Minnneapolis, MN, USA); cellular senescence (SA-$\beta$-galactosidase$^+$; catalog. no. B4252-100MG; Sigma-Aldrich; catalog. no. 12767; Cell Signaling, Danvers, MA, USA); CD68$^+$ CD206$^-$ (catalog. no. 14-068-82; Thermo Fisher Scientific, MA, USA) and CD68$^+$ (catalog. no. 14-068-82; Thermo Fisher Scientific) CD206$^+$ (catalog. no. AF2534; R&D Systems) macrophages; senescent macrophages (CD11b$^+$; catalog. no. MON1019-1; Cell Sciences, Newburyport, MA, USA; p21$^+$; catalog. no. ab109199; Abcam); Neutrophils (MPO$^+$; catalog. no. GTX135125; GeneTex, Irvine, CA, USA; CD66b$^+$, catalog. no. MAG-51806; Thermo Fisher Scientific); and dystrophin (dystrophin$^+$; catalog. no. AB15277; Abcam). Full details of the immunofluorescence protocols are provided in the Supporting Information (Doc. A1) and previous work (Reidy et al., 2019a; Fix et al., 2021; Petrocelli et al., 2021; Petrocelli et al., 2023). Images were acquired using a Zeiss Axioscan 7 fluorescence microscope (Zeiss, Jena, Germany) attached with an X-Cite 120 LED Boost illumination system (Excelitas Technologies, Waltham, MA, USA) with a 10× or 20× objective.

### Image analyses

One entire section was analyzed for each time point for each participant, and areas of tissue folding along with disrupted or elongated fibres were excluded from analyses. MuscleJ automated analyses was used to identify MHC I and IIA fibres and myofibre cross-sectional area (CSA) along with determining myonuclear number and central myonucleated fibres. Immune cell content, FAPs, cellular senescence, CD68$^+$ CD206$^-$ and CD68$^+$ CD206$^+$ macrophages, senescent macrophages, and neutrophils were determined by assessing the area of the entire sections (excluding folds and disrupted/elongated fibres) and counting each of the cell types and dividing the number of cells by the total area. For assessing collagen content, we assessed three random 400 × 400 μm$^2$ squares and set thresholds that mask the collagen content using FIJI (NIH, Bethesda, MD, USA) and determined the percent area of Sirius Red. Distropohin$^-$ fibres were determined by counting all the viable fibres and the number of fibres with missing dystrophin and divided the number of dystrophin$^-$ fibres by the total number of fibres.

### Statistical analysis

Statistical analyses were performed using Prism, version 9 (GraphPad Software Inc., San Diego, CA, USA). Two-way ANOVA was used to determine main effects while Sidak's post-hoc analysis was utilized for time-by-age interactions. If missing data points were present or groups were unbalanced (one YA and OA are missing a biopsy at one of the recovery timepoints), mixed effects analysis was used in place of ANOVA. Data sets were assessed visually for normality via Q-Q plots, skewedness and kurtosis, and tested with the Shapiro–Wilk test if necessary. Participants were removed from the histological analyses as a result of the exaggerated responses that were considered an outlier via Grubbs' test (two-sided, $\alpha < 0.05$). These specifications were established prior to viewing the data. Pearson correlations were used to determine associations between dependent measurements and were controlled for false discovery rate ($q < 0.05$). $P < 0.05$ was considered statistically significant and data are reported as the mean ± SD.

## Results

### Physical activity and lean mass

There was a main effect for age ($P < 0.001$) and time ($P < 0.001$) such that both young and OAs reduced their step count by ∼50% from PRE to post-immobilization which then returned to near baseline levels by 7 days of recovery (see Supporting Information, Table A1). Additionally, there was a time-by-age interaction ($P = 0.021$) in which the YA had a greater step count than OA at PRE ($P < 0.001$), post-immobilization ($P = 0.023$) and at 7 days of recovery ($P = 0.002$). There was a main effect of age and time in the immobilized leg (age: $P = 0.012$; time: $P < 0.001$) and quadriceps (age: $P = 0.010$; time: $P < 0.001$) and combined right and left leg (age: $P = 0.008$; time: $P < 0.001$) and quadriceps

(age: $P = 0.006$; time: $P < 0.001$), with both groups demonstrating lower lean mass from PRE in each lean mass measurement. There were no age-by-time interactions (see Supporting Information, Table A1).

A comparison of physical activity and lean mass by age group, separated by males and females, is provided in the Supporting Information (Table A2). For step counts, both males and females demonstrated main effects of age (males: $P = 0.018$; females: $P = 0.002$) and time (males: $P < 0.001$; females: $P < 0.001$) when comparing by age groups. Regarding lean mass, young and older males exhibited main effects of age and time in the immobilized leg (age: $P = 0.003$; time: $P < 0.001$) and quadriceps (age: $P = 0.002$; time: $P = 0.001$) and combined right and left leg (age: $P = 0.002$; time: $P < 0.001$) and quadriceps (age: $P < 0.001$; time: $P < 0.001$). Similarly, females demonstrated main effects of age and time in the immobilized leg (age: $P = 0.015$; time: $P = 0.001$), combined left and right leg (age: $P = 0.001$; time: $P = 0.003$), and combined left and right quadriceps (age: $P = 0.036$; time: $P = 0.010$). For immobilized quadriceps lean mass, only a main effect of age was observed in the females ($P = 0.018$).

### Myofibre measurements

We also examined myofibre characteristics with histochemistry following immobilization and recovery (see Supporting Information, Table A3). There were no age, time, or time-by-age interactions in myosin heavy chain (MHC) composition. Additionally, there were no age or time-by-age interactions in MHC I myofibre CSA. However, there was a time effect for MHC I CSA ($P = 0.038$) in which there was a decreased myofibre CSA from PRE. MHC IIA myofibre CSA did not exhibit a time or time-by-age interaction but had a main effect of age ($P < 0.001$). Regarding myonuclear number, there were no time or age main effects, but there was a time-by-age interaction ($P = 0.026$). Concerning myonuclear domain (MND), there was no time-by-age interaction but there were a main effect of age ($P < 0.001$) and time ($P = 0.006$) as both groups combined decreased MND from PRE. This reduction in cytoplasmic volume per nucleus, despite no net change in their myonuclear number, support a decreased myofibre size following immobilization.

A summary of myofibre characteristics by age group, stratified by sex, is provided in the Supporting Information (Table A4). Among males, no main effects of age or time, nor age-by-time interactions, were observed for MHC composition or MHC I fibre CSA. However, a main effect of age was detected for MHC IIA fibre CSA ($P = 0.002$). No main effects or interactions were observed for myonuclear number or MND between the young and older males. Similarly, among females, there

were no main effects of age or time, nor age-by-time interactions, for MHC composition or MHC I fibre CSA. By contrast, MHC IIA fibre CSA demonstrated main effects of age ($P < 0.001$) and time ($P < 0.001$), as well as an age-by-time interaction ($P < 0.001$). Younger females exhibited greater MHC IIA CSA at PRE ($P = 0.001$), post-immobilization ($P = 0.016$) and at 2 days of recovery ($P = 0.013$) with a trend toward a difference at 7 days of recovery ($P = 0.073$). No main effects or interactions were observed for myonuclear number or MND between young and older females.

### Aged muscle exhibited evidence of muscle regeneration during recovery

There were modest signs of muscle regeneration in spite of no overt fibre damage or necrosis (see Supporting Information, Fig. A1). Although the average number of central myonuclei per fibre (CMN/f) did not differ between groups at PRE, OA displayed a greater increase in CMN/f at 7 days recovery than YA ($P = 0.009$) (Fig. 2*A–C*), suggesting that the OA muscle underwent a level of muscle regeneration compared to the YA following immobilization. There were no differences between the young and older males in CMN/f at any time point. In contrast, older females exhibited a greater change in CMN/f from PRE to 2 days recovery than younger females ($P < 0.001$), whereas changes in CMN/f were similar between age groups at 7 days recovery (see Supporting Information, Fig. A2*A–C*).

Due to this evidence of regeneration, we explored signs of muscle damage by examining tissue inflammation as defined as immune cellular infiltration within the skeletal muscle via haematoxylin and eosin (H&E) staining (Fig. 2*D–F*). Though the cellular density was similar between groups at PRE, a substantial increase in cellular infiltration was observed in OA at 7 days recovery compared to YA ($P = 0.002$), suggesting evidence of a heightened muscle immune cell response within a week following immobilization in OA. There were no statistical differences between young and older males in immune cell infiltration at any time point. In contrast, older females exhibited greater immune cell infiltration relative to PRE at post-immobilization ($P = 0.008$), 2 days recovery ($P = 0.003$), and 7 days recovery ($P = 0.002$) than young females (see Supporting Information, Fig. A2*D–F*). These findings were confirmed by staining for CD45$^+$ cells (see Supporting Information, Fig. A2*G–I*).

### Macrophage content increases in aged muscle during recovery

Given the increased immune cell infiltration in OA during recovery, immunofluorescence was used to phenotype

immune cell involved with muscle repair and remodelling. We first quantified neutrophils (MPO$^+$ cells) across time and age groups but found no notable differences between YA and OA across all time points (see Supporting Information, Fig. A3$A$–$C$). Similarly, there were no sex-specific differences in neutrophil infiltration at any time point (see Supporting Information, Fig. A3$D$–$F$). The minimal changes in neutrophil content were confirmed by staining for CD66b (see Supporting Information, Fig. A3$G$–$I$). We then turned our attention on macrophage content and inflammatory states, recognizing their critical role in inflammation and tissue remodelling (Dort et al., 2019; Peck et al., 2022).

At PRE, there were no differences between YA and OA in total (CD68$^+$ cells), CD68$^+$ CD206$^-$ and CD68$^+$ CD206$^+$ macrophage densities. However, OA demonstrated a robust total macrophage response following immobilization ($P < 0.001$) and at 7 days

recovery ($P = 0.001$) compared to YA (Fig. 3$A$ and $B$). Specifically, the OA CD68$^+$ CD206$^-$ macrophage density increased at post-immobilization ($P = 0.001$) (Fig. 3$C$ and $D$), while CD68$^+$ CD206$^+$ macrophage content was higher at post-immobilization ($P = 0.001$) and at 7 days of recovery compared to YA ($P < 0.001$) (Fig. 3$E$ and $F$). There were no statistical differences between young and older males in total, CD68$^+$ CD206$^-$, and CD68$^+$ CD206$^+$ macrophage infiltration at any time point. In contrast, older females exhibited greater macrophage infiltration than young females, with increases in total macrophages at post immobilization ($P = 0.007$) and 7 days of recovery ($P = 0.003$) relative to PRE, and in CD68$^+$ CD206$^+$ macrophages at 7 days of recovery ($P = 0.008$). No differences were observed in CD68$^+$ CD206$^-$ macrophage infiltration between young and older females at any time point (see Supporting Information, Fig. A4$A$–$I$). Both immune cell infiltration

## Central Myonucleated Fibers

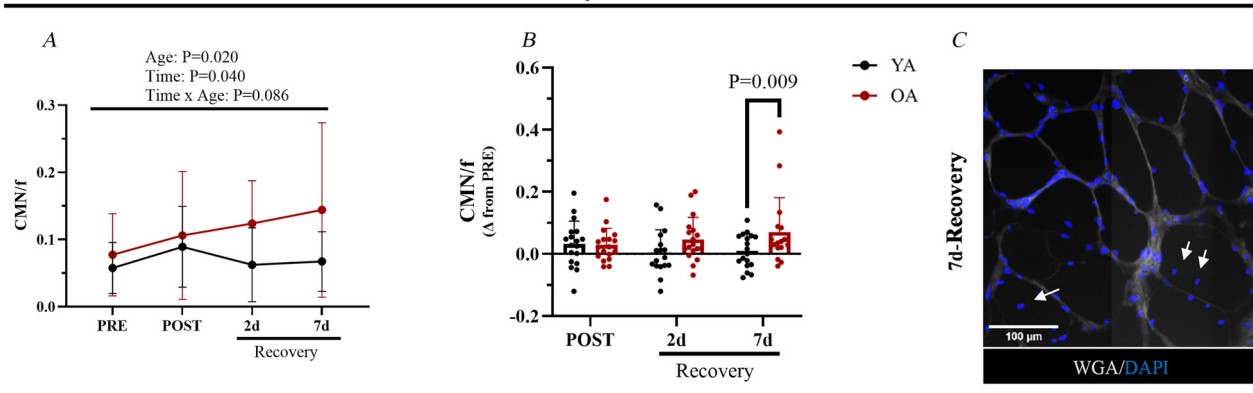

## Immune Cell Infiltration

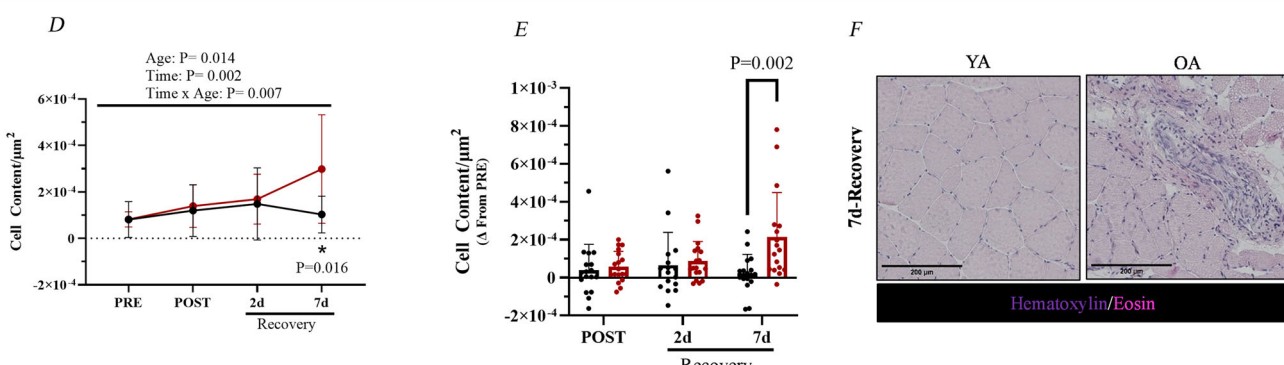

**Figure 2. Muscle regeneration and immune cell infiltration**
*A*–*C*, muscle regeneration. *A*, central myonuclei per fibre (CMN/f) (DAPI$^+$) (PRE: YA: *n* = 18; OA: *n* = 18; POST: YA: *n* = 18; OA: *n* = 18; 2 days of recovery: YA: *n* = 18; OA: *n* = 17; 7 days of recovery: YA: *n* = 18; OA: *n* = 17). *B*, change in CMN/f (Δ) from PRE. *C*, representative images of CMN fibres at 7 days of recovery. *D*–*F*, immune cell infiltration. *D*, cellular infiltration per µm$^2$ (PRE: YA: *n* = 17; OA: *n* = 18; POST: YA: *n* = 17; OA: *n* = 18; 2 days of recovery: YA: *n* = 16; OA: *n* = 18; 7 days of recovery: YA: *n* = 17; OA: *n* = 17. *E*, change in cellular infiltration per µm$^2$ (Δ) from PRE. *F*, representative images of cellular infiltration via H&E staining at 7 days of recovery. Data are shown as the mean ± SD. OA, older adults; YA, young adults

assays (H&E: $r = 0.67$, $P < 0.001$; CD45: $r = 0.73$, $P < 0.001$) (Fig. 3*H*) were significantly associated with the total macrophage infiltration (CD68$^+$ cells) suggesting that most of the immune cell were indeed macrophages.

Next, we explored FAP (PDGFR$\alpha^+$ cells) and satellite cell (PAX7$^+$ cells) content in skeletal muscle samples across the immobilization and recovery timeline because macrophages have been shown to regulate the cellular activity of these cell types during muscle recovery (Ahmadi et al., 2022; Brorson et al., 2025). Despite the increased macrophage infiltration in OA during recovery, we found little changes in FAP density within the histological sections that would suggest differential changes between age groups (Fig. 4*A–C*). Similarly, satellite cell content did not change between age groups at any timepoint (Fig. 4*D–F*). There were

also no sex-specific differences in FAP (see Supporting Information, Fig. A5*A–C*) or satellite cell content (see Supporting Information, Fig. A5*D–F*) at any the time point.

## Elevated senescence, including senescent macrophages, in aged muscle during recovery

Given the notable macrophage infiltration primarily in OA at 7 days of recovery, we explored cellular senescence, and, specifically, senescent macrophages as a potential driver of altering function given that senescent cells accumulate following muscle damage in aged muscle and related to impaired muscle recovery (Dungan et al., 2022; Chen et al., 2025). At PRE, there was no difference

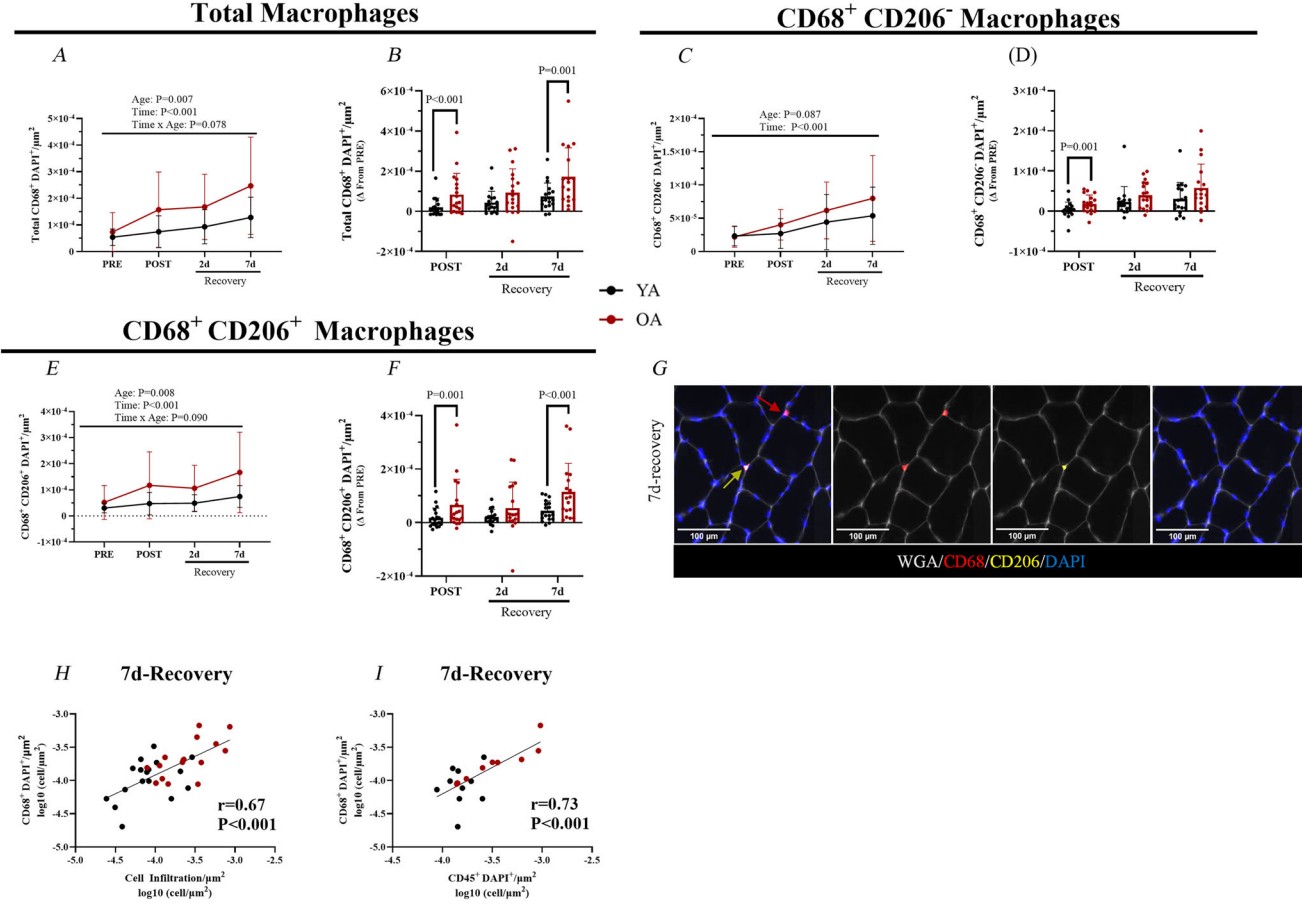

**Figure 3. Macrophage infiltration and characterization**
***A*** and ***B***, total macrophages. ***A***, total macrophages per μm$^2$ (CD68$^+$ DAPI$^+$) (PRE: YA: *n* = 18; OA: *n* = 18; POST: YA: *n* = 18; OA: *n* = 18; 2 days of recovery: YA: *n* = 17; OA: *n* = 18; 7 days of recovery: YA: *n* = 18; OA: *n* = 17). ***B***, change in total macrophages per μm$^2$ (Δ) from PRE. ***C*** and ***D***, CD68$^+$ CD206$^-$ macrophages. ***C***, CD68$^+$ CD206$^-$ macrophages per μm$^2$ (CD68$^+$ CD206$^-$ DAPI$^+$). ***D***, change in CD68$^+$ CD206$^-$ macrophages per μm$^2$ (Δ) from PRE. ***E*** and ***F***, CD68$^+$ CD206$^+$ macrophages per μm$^2$ (CD68$^+$ CD206$^+$ DAPI$^+$). ***F***, change in CD68$^+$ CD206$^+$ macrophages per μm$^2$ (Δ) from PRE. ***G***, representative image: Immunofluorescence images from 7 days of recovery to identify CD68$^+$ (total), CD68$^+$ CD206$^-$, CD68$^+$ CD206$^+$ macrophages. ***H*** and ***I***, correlations. ***H***, correlation between immune cell infiltration (H&E) and macrophages (CD68$^+$ DAPI$^+$) at 7 days of recovery. ***I***, correlation between immune cell infiltration (CD45$^+$ DAPI$^+$) and macrophages (CD68$^+$ DAPI$^+$) at 7 days of recovery. Data are shown as the mean ± SD. MACs, macrophages; OA, older adults; YA, young adults.

between YA and OA in broad cellular senescence (SA-$\beta$-galactosidase$^+$ cells) or senescent macrophages (CD11b$^+$ p21$^+$ cells; $P = 0.055$) content at baseline. However, an age-related difference emerged during recovery such that OA exhibited a greater increase in overall content of cellular senescent cells compared to YA at 2 days ($P < 0.001$) and 7 days of recovery ($P = 0.002$) (Fig. 5$A$–$C$). Macrophage infiltration (CD68$^+$ DAPI$^+$ cells) were significantly associated with the total number of senescent cells (SA-$\beta$-Gal$^+$ cells) at 7 days of recovery ($r = 0.68$, $P < 0.001$) (Fig. 5$D$), suggesting a link between macrophage infiltration and cellular senescence. This was mirrored by a similar increase in senescent macrophage cell density at 7 days of recovery in OA compared to YA ($P < 0.001$) (Fig. 5$E$–$G$).

There were no statistical differences in cellular senescence between young and older males at any time point. By contrast, older females exhibited greater cellular senesce at 7 days of recovery relative to PRE compared to young females ($P = 0.009$) (see Supporting Information, Fig. A6$A$–$C$). Additionally, when stratified by sex, no differences were observed between young and older individuals in senescent macrophage content at any time point (see Supporting Information, Fig. A6$D$–$F$).

## Muscle collagen content increases in the OAs during recovery

Because of the robust macrophage response and expansion of the senescent macrophage population in OA during muscle recovery, we assessed collagen content in skeletal muscle sections using the Sirius Red histochemical assay. This focus was driven by the established role of macrophages in ECM remodelling (Abramowitz et al., 2018). Though OA had more collagen content than YA at PRE ($P = 0.015$), the collagen accumulation accrued in the OA at 7 days (*vs.* PRE)

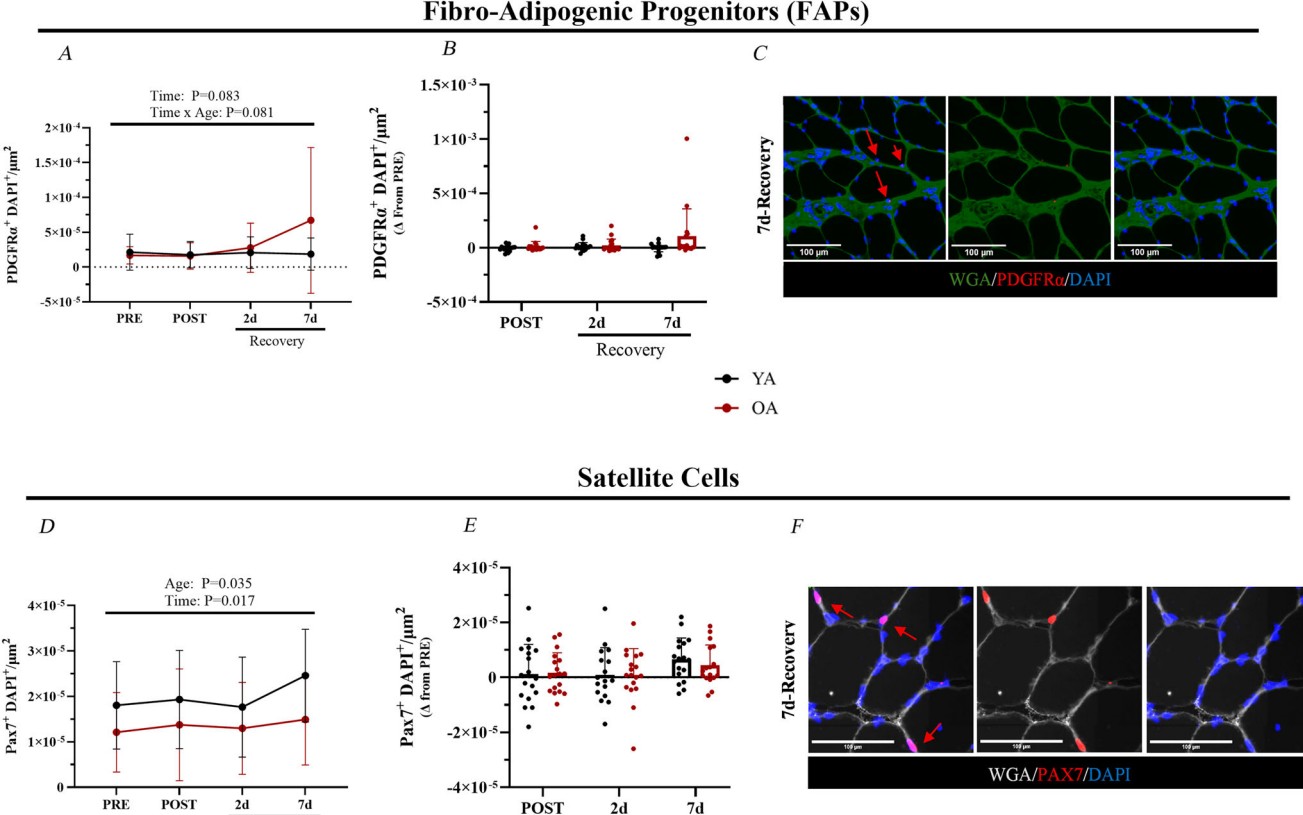

**Figure 4. Fibro-adipogenic progenitor and satellite cell content**

**A–E**, fibro-adipogenic progenitors. **A**, fibro-adipogenic progenitors (FAPs) per µm$^2$ (PDGFR$\alpha^+$ DAPI) (PRE: YA: $n = 18$; OA: $n = 18$; POST: YA: $n = 18$; OA: $n = 18$; 2 days of recovery: YA: $n = 17$; OA: $n = 18$; 7 days of recovery: YA: $n = 18$; OA: $n = 17$). **B**, change in FAPs per µm$^2$ ($\Delta$) from PRE. **C**, representative images used to identify FAPs$^+$ cells at 7 days of recovery. **D–F**, satellite cells. **D**, satellite cells per µm$^2$ (Pax7$^+$ DAPI$^+$) (PRE: YA: $n = 18$; OA: $n = 18$; POST: YA: $n = 18$; OA: $n = 18$; 2 days of recovery: YA: $n = 17$; OA: $n = 18$; 7 days of recovery: YA: $n = 18$; OA: $n = 17$). **E**, change in satellite cells per µm$^2$ ($\Delta$) from PRE. **F**, representative images used to identify satellite$^+$ cells at 7 days of recovery. Data are shown as the mean ± SD.

## Senescent Cells

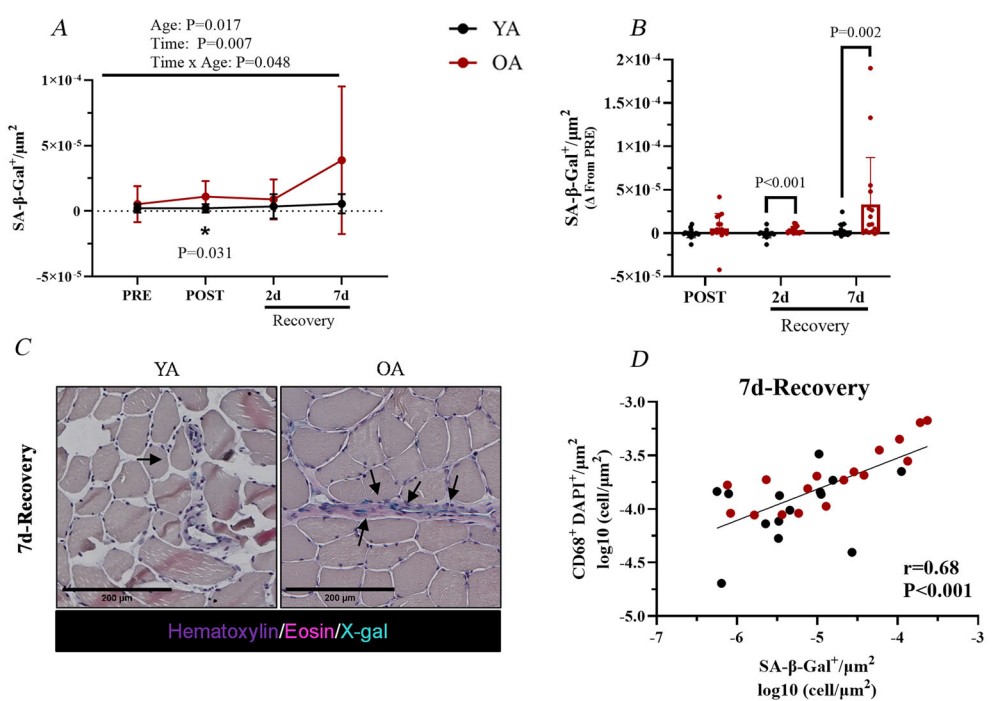

## Senescent Macrophages

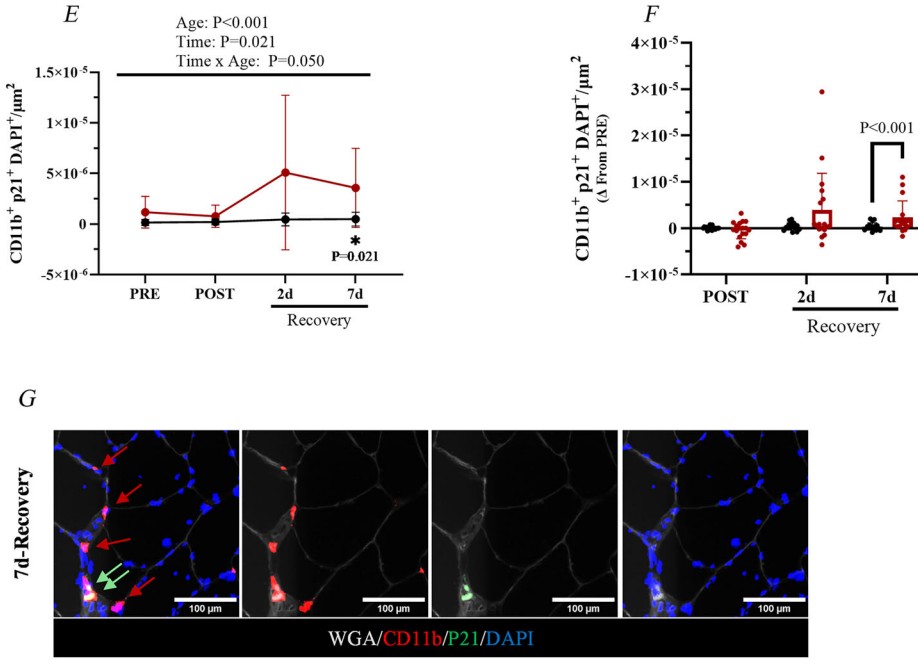

**Figure 5. Cellular senescence and senescent macrophages**

*A–D*, cellular senescence. *A*, senescent cells per $\mu m^2$ (SA-$\beta$-Gal$^+$) (PRE: YA: $n = 17$; OA: $n = 17$; POST: YA: $n = 17$; OA: $n = 17$; 2 days of recovery: YA: $n = 16$; OA: $n = 17$; 7 days of recovery: YA: $n = 17$; OA: $n = 16$). *B*, change in senescent cells per $\mu m^2$ ($\Delta$) from PRE. *C*, representative images used to identify SA-$\beta$-Gal$^+$ cells at 7 days of recovery. *D*, correlation between macrophages (CD68$^+$ DAPI$^+$ cells) and senescent cells (SA-$\beta$-Gal$^+$) at 7 days of recovery. *E–G*, senescent macrophages. *E*, senescent macrophages per $\mu m^2$ (CD11b$^+$ p21$^+$ DAPI$^+$) (PRE: YA:

*n* = 18; OA: *n* = 18; POST: YA: *n* = 18; OA: *n* = 18; 2 days of recovery: YA: *n* = 17; OA: *n* = 18; 7 days of recovery: YA: *n* = 18; OA: *n* = 17). **F**, change in senescent macrophages per $\mu m^2$ ($\Delta$) from PRE. **G**, representative images used to identify $CD11b^+$ $p21^+$ cells at 7 days of recovery.
Data are shown as the mean ± SD. OA, older adults; YA, young adults.

compared to YA ($P < 0.001$) (Fig. 6A–C). Additionally, both older males ($P = 0.001$) and older females ($P < 0.001$) exhibited greater collagen deposition at 7 days of recovery relative to PRE compared to their young counterparts (see Supporting Information, Fig. A7A–C). Consistent with the elevated collagen accumulation in OA at 7 days of recovery, we found a moderately strong negative association between Sirius Red percent area and myofibre CSA ($R = -0.49$, $P = 0.004$) (Fig. 6D), suggesting that increased collagen deposition is linked to lower myofibre size during muscle recovery. Furthermore, we found a moderately strong association between Sirius Red percent area and cellular senescence ($R = 0.39$, $P = 0.035$) (Fig. 6E) and senescent macrophage density ($R = 0.69$, $P < 0.001$) (Fig. 6F) at 7 days of recovery.

## Discussion

In this clinical trial, we examined muscle interstitial cell dynamics during the first week of ambulatory recovery following 14 days of immobilization in young and OAs. The major findings indicate that, during recovery, OAs (*vs.* young), exhibited elevated: (i) immune cell and $CD68^+$ $CD206^+$ macrophage infiltration; (ii) expansion of cellular senescent cells (including senescent macrophages); (iii) collagen deposition; and (iv) a senescent macrophage population that positively corresponded to muscle collagen content. These data suggest that, during the early phase of remobilization, OA muscle evoke a rapid and robust immune cell infiltration and macrophages response corresponding with increased cellular senescence.

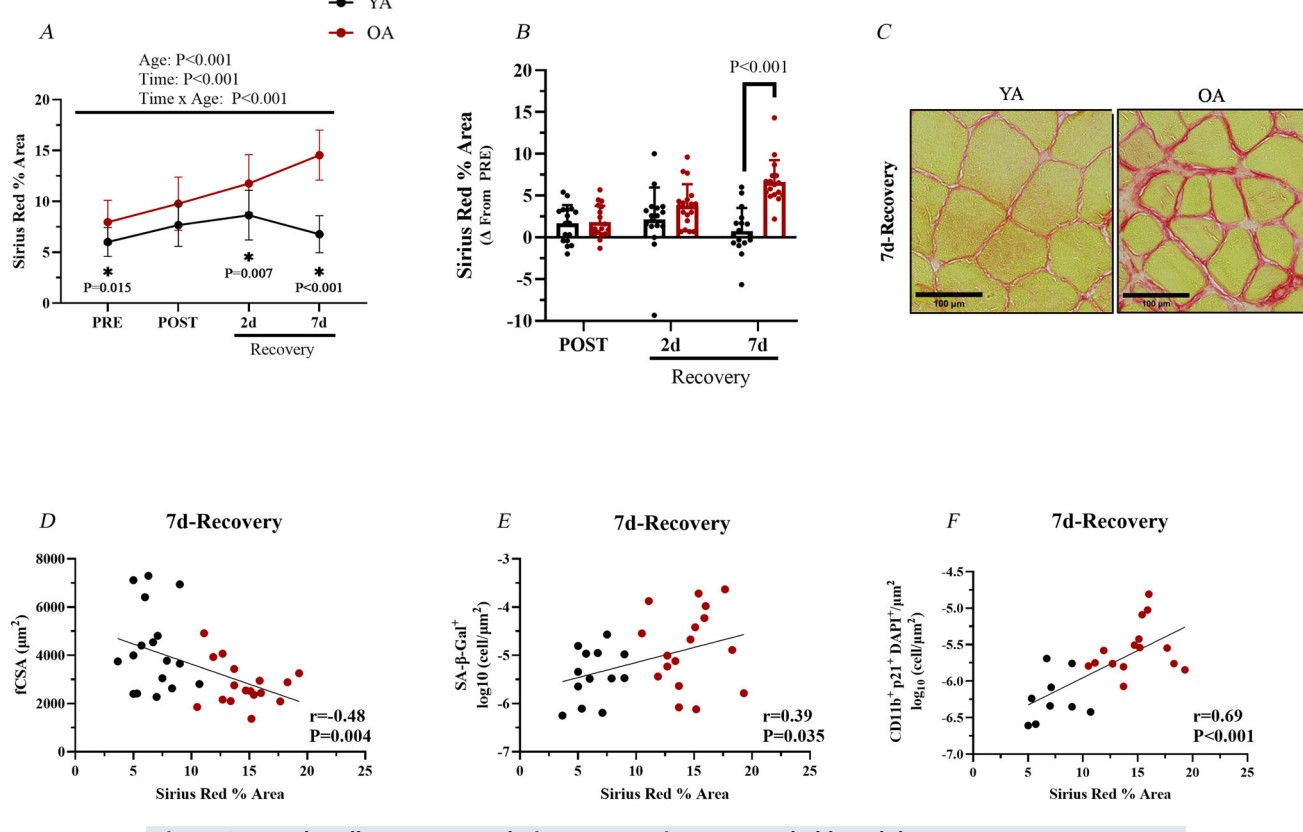

**Figure 6. Muscle collagen content during recovery in young and older adults**
**A**, Sirius Red percent area (PRE: YA: *n* = 17; OA: *n* = 18; POST: YA: *n* = 17; OA: *n* = 18; 2 days of recovery: YA: *n* = 16; OA: *n* = 18; 7 days of recovery: YA: *n* = 17; OA: *n* = 17). **B**, change in Sirius Red percent area ($\Delta$) from PRE. **C**, representative images of Sirius Red staining at 7 days of recovery. **D**, correlation between Sirius Red percent area and myofibre CSA at 7 days of recovery. **E**, correlation between Sirius Red percent area and cellular senescence. **F**, correlation between Sirius Red percent area and senescent macrophages ($CD11b^+$ $p21^+$ $DAPI^+$) at 7 days of recovery. Data are shown as the mean ± SD. MAC, macrophages; OA, older adults; YA, young adults.

Subsequently, this altered muscle macrophage response may be related to elevated collagen accumulation during muscle recovery in OAs.

The major finding of the present study was the robust cellular infiltration of immune cells, such as CD68[+] CD206[+] macrophages, that occurred in the OA muscle during recovery following 2 weeks of limb immobilization. During normal muscle repair, a timely transition from pro-inflammatory to anti-inflammatory macrophages is critical for effective regeneration (Arnold et al., 2007; Saclier et al., 2013; Muire et al., 2020; Ahmadi et al., 2022). An imbalance or overabundance of either phenotype can be detrimental to the repair process (Shireman et al., 2007; Villalta et al., 2011; Ahmadi et al., 2022). This aligns with findings from our prior work and others, where the timing of macrophage infiltration and inflammatory profile during recovery from injury or disuse is dysregulated in aged mice and humans (White et al., 2015; Sloboda et al., 2018; Reidy et al., 2019a; Sorensen et al., 2019; Fix et al., 2021; Ahmadi et al., 2022). We also found it notable that macrophages accumulated in skeletal muscle of OAs during the immobilization phase, prior to a reloading stimulus. We have previously observed this immune cell phenomenon with reduced physical activity in OAs (Reidy et al., 2019c) and this is paralleled during immobilization in mice in other studies (Kawanishi et al., 2018). We surmise that this heightened muscle macrophage content during disuse may be a result of reduced muscle contractile activity and subsequent decreased local blood flow (Tyml & Mathieu-Costello, 2001) resulting an accumulation of muscle macrophages. Interestingly, cyclical compressions during immobilization decreased muscle macrophage content suggesting a macrophage–muscle contraction mechanism (Saitou et al., 2018). We also speculate that the pronounced macrophage infiltration observed during the recovery in the OAs may reflect increased and possibly dysregulated recruitment, potentially driven by CCR2 signalling (Martinez et al., 2010; Guo et al., 2024). However, further studies are warranted to investigate what may be driving robust immune cell infiltration in aged muscle.

Although muscle damage and regeneration typically accompany local trauma or vigorous exercise (Pizza & Buckley, 2023), the OAs (compared to young) in the present study had an elevated number of central myo-nucleated fibres, inferring a muscle regeneration response during recovery from immobilization. This suggests that the reloading phase in aged muscle induced a stress signal, akin to that seen after muscle damage or strenuous exercise (Murach et al., 2018; Goh et al., 2019), which initiated a cascade of cellular events for muscle repair, including a prominent immune cell infiltration response, particularly involving macrophages. Interestingly, aged muscle is more sensitive to damaging stimuli and recovery from disuse atrophy (Kanazawa et al., 2017; Kanazawa et al., 2021), perhaps accounting for the aged-related differences in damage and regeneration seen between these two age cohorts. We acknowledge that this regeneration response was independent of changes in satellite cell content (Sorensen et al., 2019; Ahmadi et al., 2022), but could have been a factor of a missed time point in this experimental design or lack of specifying satellite cell activation. Similarly, it is also possible that we missed a transient neutrophil infiltration and clearance as a result of a missed time point immediately after immobilization. Together, these data suggest reloaded OA muscle is characterized by a heightened immune cell infiltration response, with much of these resembling CD68[+] CD206[+] macrophages.

Our data further demonstrate a significant increase in overall muscle cellular senescence and senescent macrophages in OAs during recovery. Given that immunosenescence is prevalent with ageing (Yousefzadeh et al., 2021; Moiseeva et al., 2023), the robust cell infiltration may have triggered an expansion of senescent macrophages in aged skeletal muscle tissue. An accumulation of muscle cellular senescence has been observed with ageing (Zhang et al., 2022) and in aged mice following injury and functional overload corresponding to impaired muscle remodelling (Dungan et al., 2022; Zhang et al., 2022). Although senescence can occur in diverse cell types, the prevalence and specific consequences of macrophage senescence are becoming more recognized (Chen et al., 2013). Senescent macrophages are reported to have compromised functions, including impaired clearance of abnormal cells and misfolded proteins (Lian et al., 2020; Wang et al., 2024). This functional impairment is particularly relevant given findings that *in vitro* senescent macrophages contribute to lung fibrosis (Su et al., 2021). This senescent macrophage burden could directly hinder critical macrophage functions such as phagocytosis and effective ECM remodelling, thereby contributing to the observed excessive collagen accumulation within OA skeletal muscle after disuse.

Finally, it is noteworthy to mention that we observed little muscle myofibre changes in young and OAs following immobilization and early recovery. Despite limb immobilization being a potent model of muscle atrophy, there are discrepant findings in myofibre size using this model (Yasuda et al., 2005; Snijders et al., 2014; Franchi et al., 2025; Michel et al., 2025), with even fewer studies conducting immobilization in OAs (D'Antona et al., 2003; Suetta et al., 2009; Suetta et al., 2012). Outside the appreciated limitation of myofibre size methodology (Michel et al., 2025), we consider that the lack of myofibre changes, particularly in the OAs, could be partially confounded by the robust macrophage infiltration during the immobilization and initial muscle reloading period. Previous work has established that

exercise-induced muscle damage induces an increased macrophage content (Sorensen et al., 2019; Ahmadi et al., 2022) and that macrophage infiltration is associated with muscle swelling (Smith, 1991; Kristiansen et al., 2014). Indeed, rats during early recovery following hind-limb unloading were characterized with a transient increase in muscle mass (Baehr et al., 2016). Thus, it is possible that immobilization and early recovery induce a robust macrophage accumulation response and associated muscle swelling, thereby masking observable changes in muscle size at the myofibre level.

A limitation of the present study is that menstrual cycles phases were not controlled among the young females in the YA cohort. Oestrogen levels fluctuate substantially across the menstrual cycle, increasing by as much as 10–50 fold (Pfeifer et al., 2024). Oestradiol has been shown to exert anti-inflammatory effects and attenuates macrophage response following muscle damage (Enns & Tiidus, 2008). Furthermore, post-menopausal women typically exhibit elevated systemic inflammation at baseline and more susceptible to autoimmune and inflammatory diseases (Abildgaard et al., 2020). However, hormone replacement therapy (estradiol alone or combined with progestin) has been associated with a blunted inflammatory response after exercise-induced muscle damage om this population (Dieli-Conwright et al., 2009). The differences in sex hormones between the young and older females may have partly contributed to the heightened immune cell responses observed collectively in the OA cohort and merit further investigation.

In summary, our findings demonstrate that muscle reloading following disuse induces a distinct mechanical stress response in OAs characterized by a robust infiltration of immune cells, notably macrophages. This response is marked by an imbalance predominance of CD68$^+$ CD206$^-$ macrophages and a concurrent over-accumulation of senescent cells, including senescent macrophages, which collectively appear to perturb ECM remodelling. We propose that this altered immune cell response and senescent burden hinders critical macrophage function during recovery in aged muscle.

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

# Additional information

## Data availability statement

Data produced in this study and detailed protocols are available in the Supporting Information. Data and protocol are will also be made available upon request.

## Competing interests

The authors declare that they have no competing interests.

## Author contributions

M.J.D. and R.M.O'C conceived and designed the research. C.M.S. was responsible for experiments and data analysis. C.M.S., Z.J.F., P-EB, E.M.Y., R.J.C. and M.J.D. analyzed and interpretated the results of experiments. C.M.S. and M.J.D. were responsible for manuscript preparation. All authors were responsible for editing the final manuscript.

## Funding

Funding for this project was provided by R01AG076075 (MD and RO). PEB (AHA 25PRE1362056), ZF (AHA 26POST1548675) and EY (NIH 5T32DK091317) were supported with training fellowships. The content is solely the responsibility of the authors and does not necessarily represent the official views of the National Institute of Health.

## Acknowledgements

We thank the participants for their dedication and effort, and the clinical and Translational Science Institute nursing and medical staff for assistance with muscle biopsies, blood draws, and patient care. We also thank Huntsman Cancer Institute Cell Imaging Core for their experimental expertise. The graphical abstract and trial schematic were created with Biorender.com.

## Keywords

ageing, disuse atrophy, fibrosis, immunosenescence, inflammation

## Supporting information

Additional supporting information can be found online in the Supporting Information section at the end of the HTML view of the article. Supporting information files available:

**Peer Review History**
**Supporting Information**

