## [Peer Review History · The Journal of Physiology]

Robust skeletal muscle immune cell infiltration and macrophage senescence occur within a week of recovery after limb immobilization in older adults

Chad M Skiles, Zachary James Fennel, Paul-Emile Bourrant, Elena M Yee, Robert J Castro, Hannah A Zabriskie, Melina Itinose, Mark A Supiano, Ryan O'Connell, and Micah J Drummond

DOI: 10.1113/JP290346

Corresponding author(s): Micah Drummond (micah.drummond@hsc.utah.edu)

The following individual(s) involved in review of this submission have agreed to reveal their identity: Caíque Figueiredo (Referee #1)

Review Timeline:

Submission Date:	16-Oct-2025
Editorial Decision:	18-Nov-2025
Revision Received:	27-Jan-2026
Accepted:	25-Feb-2026

Senior Editor: Karyn Hamilton

Reviewing Editor: Nima Gharahdaghi

Transaction Report:

Re: JP-RP-2025-290346 **"Robust immune cell infiltration and macrophage senescence occurs during early recovery following limb immobilization in older adult skeletal muscle"** by Chad M Skiles, Zachary James Fennel, Paul-Emile Bourrant, Elena M Yee, Robert J Castro, Hannah A Zabriskie, Melina Itinose, Mark A Supiano, Ryan O'Connell, and Micah Drummond

Dear Dr Drummond,

Thank you for submitting your manuscript to The Journal of Physiology. It has been assessed by a Reviewing Editor and by 2 expert referees and we are pleased to tell you that it is potentially acceptable for publication following satisfactory major revision.

Please address all the points raised and incorporate all requested revisions or explain in your Response to Referees why a change has not been made. We hope you will find the comments helpful and that you will be able to return your revised manuscript within 2 months. If your article is NOT for a Special Issue, you may have 9 months to revise. If you require an extension, please contact journal staff: jp@physoc.org. Please note that this letter does not constitute a guarantee for acceptance of your revised manuscript.

REVISION CHECKLIST:

Upload a full Response to Referees file. To create your 'Response to Referees': copy all the reports, including any comments from the Senior and Reviewing Editors, into a Microsoft Word, or similar, file and respond to each point, using

font or background colour to distinguish comments and responses and upload as the required file type.

We look forward to receiving your revised submission.

Yours sincerely,

Karyn Hamilton
Senior Editor
The Journal of Physiology

EDITOR COMMENTS

Reviewing Editor:

Comments for Authors to ensure the paper complies with the Statistics Policy:

Statistical reporting is incomplete (assumptions, outlier policy, multiple testing control) and must be explicit. Sex as a biological variable is underaddressed; at minimum the authors should provide sex-disaggregated data and acknowledge menstrual phase as an uncontrolled factor in young women. The PBMC/macrophage collagenolysis experiment is underpowered and selectively sampled; without added controls (assay background definition, differentiation checks, polarization validation, and, ideally, senescence markers in vitro) it should be framed as exploratory and not used to support causal inferences. Sampling rigor and image-analysis details need to be fully specified

Comments to the Author:

- The study addresses a clinically relevant problem with a solid human design and clear age-related differences in inflammation and collagen remodeling during reloading. The central message is promising, but the manuscript requires substantial methodological tightening, clearer phenotype definitions, fuller statistical transparency, and additional analyses to align with the standards of The Journal of Physiology.
- The macrophage phenotyping needs greater specificity or more cautious terminology. As Referee #1 notes, CD68 is pan-myeloid, and CD206 alone does not define an alternatively activated phenotype. Either expand the panel to include at least one additional marker for each state, such as HLA-DR or CD86 for classically activated cells and CD163 for alternatively activated cells, or relabel all instances to neutral descriptors ("CD68⁺CD206⁻ macrophages" and "CD68⁺CD206⁺ macrophages") and temper any mechanistic language linked to "M1/M2". Similarly, in tissue senescence analyses, you identify "senescent macrophages" by CD11b with p21. CD11b is not macrophage-specific in human tissue; either add CD68 to the co-stain or consistently relabel this compartment as "senescent myeloid cells," including in the abstract and conclusions. This point aligns with Referee #2's concern regarding Fig. 5.
- The time-course of "total infiltration" versus lineage-defined counts must be reconciled. Referee #2 points out that the H&E-based infiltration curve does not map cleanly to macrophage accrual in young adults and that neutrophil responses appear surprisingly small. Validate neutrophil detection with a human-appropriate marker such as CD66b or neutrophil elastase in addition to MPO, and confirm that staining quality and quantification are robust. It seems adding a pan-leukocyte index (for example, CD45⁺ cells per area) across all time points to harmonize H&E "infiltration" with immune-lineage counts would be helpful, and clarify which non-macrophage populations account for the early post-reloading signal in young adults if macrophages do not.
- The age bias in late inflammation requires either additional data or tighter interpretation. Please provide evidence for increased recruitment at day 7 in older adults. If tissue permits, include a recruitment-associated marker such as CCR2 on CD68⁺ cells at day 7; if not feasible, state this limitation and frame late recruitment as a hypothesis. If you propose impaired immunological tolerance, provide an exploratory FOXP3⁺ Treg count in the tissue or clearly flag this as speculation and add appropriate citations. The discussion should articulate a mechanistic model that is consistent with your human data while acknowledging where evidence is currently associative.

- The collagenolysis experiment needs controls, method clarity, and reframed claims. Please define all assay controls explicitly, explain how negative values arise in the DQ-collagen readout, and report background subtraction procedures. Validate monocyte-to-macrophage differentiation and polarization in human cells with simple phenotypic checks (for example, CD14/CD68 upshift; MRC1/CD206, HLA-DR or TNF expression) and report differentiation efficiency by donor age. The donor selection strategy for this assay is biased toward "high responders"? If so, it is necessary to state that it is exploratory and show all individual points, report effect sizes with confidence intervals, and soften causal language. If possible, add p21 or SA- β -Gal to the cultured macrophages to support the proposed link between myeloid senescence and reduced collagen degradation in older adults. These adjustments address Referee #2's Major Comment on Fig. 6.
- Statistical reporting and outlier handling must be explicit. Confirm that assumptions for ANOVA, t-tests, and Pearson correlations were tested for all variables, state the diagnostics used, and describe remedies where violated, including any transformations or non-parametric alternatives. Specify the Grubbs' test parameters used for outlier removal, including alpha and whether one- or two-sided, and state whether the rule was pre-specified before viewing the data. Where you present many correlations, indicate whether you controlled the false discovery rate and, if not, provide a rationale. In tables, please denote significant main effects as well as interactions, as suggested by Referee #2.
- Sex as a biological variable needs to be handled transparently. Please acknowledge established sex differences in innate and adaptive immunity, and then discuss how these might influence your outcomes. Whether the menstrual cycle phase was controlled in young female participants; if not, state this plainly as a limitation and explain the potential impact on inflammatory readouts.
- Sampling rigor and image analysis need more detail. Report the number of sections or areas quantified per biopsy for each stain, the region-of-interest selection rules, the analyzed area per section, and any exclusion criteria for fields. Please make sure all figure axes and legends include units and normalization. For the large variance in older adults at day 2 in Fig. S1A-B, please follow Referee #2's suggestion: stratify participants into high versus low dystrophin-negative responders at day 2 and test whether this stratification predicts day-7 macrophage accrual, myeloid-cell senescence, and collagen content. Report the results regardless of outcome and comment on whether this explains variance in later endpoints.
- The PBMC culture protocol needs to state cell composition and purity. Clarify whether lymphocytes were removed before or after seeding. If not removed, explain how the culture conditions favored monocyte-derived macrophages (for example, M-CSF reliance and adherence selection) and provide a brief purity estimate or marker check.
- Textual accuracy and framing should be corrected throughout. Where the manuscript currently states that age differences emerge "in the early days following immobilization," revise to reflect that the clearest divergence occurs late in the reloading week, consistent with your data. Replace any causal phrasing that implies disruption or impairment by a particular cell phenotype with association language unless supported by direct manipulation. Ensure that each figure legend lists the exact n per group at each time point and that normalization conventions are consistent across panels.
- Because several points hinge on marker specificity and cell identity in human tissue, I expect either expanded panels or neutral relabeling to avoid over-interpretation. Because several claims rely on correlations, I expect fuller statistical transparency, clear acknowledgment of multiple testing where relevant, and effect sizes with uncertainty, not only P-values. Because you draw mechanistic inferences from an underpowered and selectively sampled ex vivo assay, I expect the claims to be reframed as exploratory/association unless you can add the requested controls and validation.
- Please make sure the Standard deviation (SD) is reported, not SEM.

Senior Editor:

Comments for Authors to ensure the paper complies with the Statistics Policy:

If you choose to revise your manuscript, please revisit The Journal's statistic policy to ensure full compliance. For example, please clearly show variability as SD (not SE), present individual data points on the graphs, and report precise p-values in the graphs (thank you for providing them in the text, but if possible, having them in the graphs too would be great). Thank you.

Comments to the Author:

Thank you for submitting your manuscript for consideration by The Journal of Physiology. As part of the peer review process, we recruited two Referees with expertise in this field of study. In addition, you have detailed feedback from our internal review process. Overall, the feedback suggests that there is potential for your manuscript to be impactful if the concerns raised during peer review can be addressed. Therefore, we would like to invite you to respond to each critique point-by-point, with corresponding major revisions to the manuscript. Additionally, please ensure compliance with The Journal's statistics policy. Finally, you may want to consider a more informative legend to accompany your graphical abstract. We very much look forward to seeing your revised manuscript and we thank you for your interest in The Journal of Physiology!

REFEREE COMMENTS

Referee #1:

The manuscript compares the dynamics of myogenic and non-myogenic cells (macrophages, FAPs), cellular senescence, and muscle collagen content during early recovery (2 and 7 days) following 14 days of unilateral limb immobilization in young (n = 18; ~24 years) and older (n = 18; ~69 years) male and female adults. Overall, the study is well designed and clearly presented, and the topic is timely. Detailed comments are provided below.

1 - Why was no additional marker used to determine the pro-inflammatory phenotype of macrophages? For example, it is possible that within the CD68+ population there are macrophages expressing other "anti-inflammatory" markers and lacking CD206 expression (e.g., CD68+CD163+). This phenotypic complexity can be crucial for the interpretation of the results.

2 - In the PBMC culture protocol, it is not clear whether the lymphocytes were removed from the wells. Could you please clarify this point?

3 - Were the statistical assumptions for the analysis of variance, t-test, and Pearson's correlation met for all variables? If so, it is important to specify this in the manuscript.

4 - There are well-established differences in both innate and adaptive immune functions, with women generally exhibiting stronger immune responses but also a greater susceptibility to autoimmune and inflammatory diseases. In this context, it would be important to consider whether sex-based differences exist in the measurements of both immune and non-immune variables examined in the study. Given the relevance of this issue, I suggest including a dedicated section on limitations to address this point. Additionally, providing the main measurements disaggregated by sex as supplementary material would substantially strengthen the manuscript.

5 - Still regarding sex differences, was the menstrual cycle phase considered during the recruitment of the young female participants? If not, this limitation should be clearly mentioned in the manuscript.

Referee #2:

This clinical trial conducted by Skiles and colleagues was designed to compare cellular dynamics (of satellite cells, FAPs, and various subtypes of immunocytes), cellular senescence, and collagen content in human limb muscle (vastus lateralis) over a timecourse of unilateral immobilization followed by reloading. The authors uncovered excessive inflammatory responses to reloading in older adults (OA) compared to young adults (YA). In particular, macrophage accumulation was exaggerated in OA late after reloading (7d); such accumulation was positively correlated with senescence in the muscle myeloid-cell compartment as well as collagen content - both of which were amplified by reloading. These findings are consistent with the comparatively larger body of literature from preclinical models of the effects of aging on cellular responses to unloading/loading and acute injury. As older populations are commonly hospitalized due to illness or injury (and the frequency of falls is on the rise), clinical trials on disuse-associated muscle atrophy, inflammation, and fibrosis are of high importance. While the correlations presented in this work highlight avenues for future mechanistic studies, there are several concerns that should be addressed to strengthen the authors' claims and provide greater clarity.

Major comments

1) The shapes of the response curves in Fig. 2D (total infiltration), Fig. S2A (neutrophils) and Fig. 3A (macrophages) are quite different, particularly for YA. Do the authors have any insights into how to harmonize these? A few suggestions for possible explanations:

- Macrophages are the predominant cell type in muscle, so one would expect that the curves in 3A would map nicely to 2D; however, this doesn't appear to be true, at least for YA. Do the authors have any data or insights into what other cell types

could account for the greater infiltration in POST and 2d not accounted for by the macrophage compartment?

- Related to the point above, the neutrophil responses are surprisingly minor, particularly in the early recovery window. One would expect that perhaps neutrophils could harmonize the shapes of the curves in 2D and 3A, but that does not appear to be true in this case. Is it possible that the MPO staining did not work properly (the area/size and shape of staining do not look as expected)? The authors should confirm with another marker such as Ly6G.

2) The protracted inflammatory responses in muscles from OA vs YA is striking (i.e., differences on day 7 of recovery). What do the authors think is the basis of this observation? This point warrants discussion in the text and perhaps additional assays:

- Based on data from preclinical models, it's likely that there is more recruitment on d7 in OA vs YA. Do the authors have any evidence that this is true in their study? There are a few ways to address this, but one would be to add CCR2 to the panel in Fig. 3.

- Active recruitment late in recovery in OA could be due to defects in mechanisms of peripheral immunological tolerance (i.e., Tregs). Do the authors have any evidence for this?

3) There are a few important issues in Figure 5:

- CD11b alone is not sufficient to identify macrophages, as it is highly expressed by other myeloid populations too (e.g., neutrophils). The authors should either enhance their panel or change "macrophages" to "myeloid cells" when referencing these data.

- The correlations in Figure 5G,H are less convincing than co-staining with the panel used in Fig. 3.

4) The results in Fig. 6G-J are interesting and quite surprising. A few comments:

- The reviewer could not find the controls for this assay defined anywhere. Please add a definition in the methods and/or figure legend and clarify how one derives negative collagen degradation values in this fluorescent collagen assay.

- The surprise from this data comes from 1) the very small difference in collagen degradation between each macrophage state within an age group and 2) the lower degradation in OA independent of age. Please provide validation of the differentiation efficiency for each age. Also, it would strengthen the conclusion of the paper and help explain surprise #2 if the authors could show that macrophages derived from OA monocytes are more senescent in their system.

5) There is large variance within the OA day 2 group in Fig. S1A,B. Can individuals be binned as high or low responders based on dystrophin-neg. fibers at this time? If so, does the fraction of dystrophin-neg. fibers in the OA day 2 condition correlate with anything on day 7, particularly accumulation of macrophages, senescent cells, and collagen?

Additional comments

1) Please specify threshold used for Grubb's test.

2) It would be helpful to denote significant main effects (perhaps using letters) in the tables, e.g. main effect of age on MHCIIA CSA (lines 261-262).

3) The authors seem to have mistakenly said "in the early days following immobilization" on line 278 instead of "in the late phase of recovery" or something similar. The data in Fig. 2D clearly indicate a protracted inflammatory response in OA vs YA, not a difference in early accrual.

4) How many sections or areas were analyzed per biopsy per stain/analysis?

5) Fig. S1A,B lacks a legend for YA and OA groups

END OF COMMENTS

This clinical trial conducted by Skiles and colleagues was designed to compare cellular dynamics (of satellite cells, FAPs, and various subtypes of immunocytes), cellular senescence, and collagen content in human limb muscle (vastus lateralis) over a timecourse of unilateral immobilization followed by reloading. The authors uncovered excessive inflammatory responses to reloading in older adults (OA) compared to young adults (YA). In particular, macrophage accumulation was exaggerated in OA late after reloading (7d); such accumulation was positively correlated with senescence in the muscle myeloid-cell compartment as well as collagen content – both of which were amplified by reloading. These findings are consistent with the comparatively larger body of literature from preclinical models of the effects of aging on cellular responses to unloading/loading and acute injury. As older populations are commonly hospitalized due to illness or injury (and the frequency of falls is on the rise), clinical trials on disuse-associated muscle atrophy, inflammation, and fibrosis are of high importance. While the correlations presented in this work highlight avenues for future mechanistic studies, there are several concerns that should be addressed to strengthen the authors' claims and provide greater clarity.

Major comments

- 1) The shapes of the response curves in Fig. 2D (total infiltration), Fig. S2A (neutrophils) and Fig. 3A (macrophages) are quite different, particularly for YA. Do the authors have any insights into how to harmonize these? A few suggestions for possible explanations:
 - Macrophages are the predominant cell type in muscle, so one would expect that the curves in 3A would map nicely to 2D; however, this doesn't appear to be true, at least for YA. Do the authors have any data or insights into what other cell types could account for the greater infiltration in POST and 2d not accounted for by the macrophage compartment?
 - Related to the point above, the neutrophil responses are surprisingly minor, particularly in the early recovery window. One would expect that perhaps neutrophils could harmonize the shapes of the curves in 2D and 3A, but that does not appear to be true in this case. Is it possible that the MPO staining did not work properly (the area/size and shape of staining do not look as expected)? The authors should confirm with another marker such as Ly6G.
- 2) The protracted inflammatory responses in muscles from OA vs YA is striking (i.e., differences on day 7 of recovery). What do the authors think is the basis of this observation? This point warrants discussion in the text and perhaps additional assays:
 - Based on data from preclinical models, it's likely that there is more recruitment on d7 in OA vs YA. Do the authors have any evidence that this is true in their study? There are a few ways to address this, but one would be to add CCR2 to the panel in Fig. 3.
 - Active recruitment late in recovery in OA could be due to defects in mechanisms of peripheral immunological tolerance (i.e., Tregs). Do the authors have any evidence for this?
- 3) There are a few important issues in Figure 5:
 - CD11b alone is not sufficient to identify macrophages, as it is highly expressed by other myeloid populations too (e.g., neutrophils). The authors should either enhance their panel or change "macrophages" to "myeloid cells" when referencing these data.

- The correlations in Figure 5G,H are less convincing than co-staining with the panel used in Fig. 3.
- 4) The results in Fig. 6G-J are interesting and quite surprising. A few comments:
 - The reviewer could not find the controls for this assay defined anywhere. Please add a definition in the methods and/or figure legend and clarify how one derives negative collagen degradation values in this fluorescent collagen assay.
 - The surprise from this data comes from 1) the very small difference in collagen degradation between each macrophage state within an age group and 2) the lower degradation in OA independent of age. Please provide validation of the differentiation efficiency for each age. Also, it would strengthen the conclusion of the paper and help explain surprise #2 if the authors could show that macrophages derived from OA monocytes are more senescent in their system.
 - 5) There is large variance within the OA day 2 group in Fig. S1A,B. Can individuals be binned as high or low responders based on dystrophin-neg. fibers at this time? If so, does the fraction of dystrophin-neg. fibers in the OA day 2 condition correlate with anything on day 7, particularly accumulation of macrophages, senescent cells, and collagen?

Additional comments

- 1) Please specify threshold used for Grubb's test.
- 2) It would be helpful to denote significant main effects (perhaps using letters) in the tables, e.g. main effect of age on MHCIIA CSA (lines 261-262).
- 3) The authors seem to have mistakenly said "in the early days following immobilization" on line 278 instead of "in the late phase of recovery" or something similar. The data in Fig. 2D clearly indicate a protracted inflammatory response in OA vs YA, not a difference in early accrual.
- 4) How many sections or areas were analyzed per biopsy per stain/analysis?
- 5) Fig. S1A,B lacks a legend for YA and OA groups

Response to Reviewers

Manuscript # JP-RP-2025-2900346

We thank the reviewers and editor for their helpful comments. As a result, over the last two months, we have conducted several new analyses including CD45 and CD66b immunofluorescence on tissue sections to help validate and interpret the age-related cell infiltration response. We have also carefully added clarity to the methodology and interpretation of the manuscript, added sex-based comparisons, and tempered language where necessary. We believe the changes made to this revision have significantly improved the impact of this manuscript.

Reviewing Editor:

Comments for Authors to ensure the paper complies with the Statistics Policy:

Statistical reporting is incomplete (assumptions, outlier policy, multiple testing control) and must be explicit. Sex as a biological variable is underaddressed; at minimum the authors should provide sex-disaggregated data and acknowledge menstrual phase as an uncontrolled factor in young women. The PBMC/macrophage collagenolysis experiment is underpowered and selectively sampled; without added controls (assay background definition, differentiation checks, polarization validation, and, ideally, senescence markers in vitro) it should be framed as exploratory and not used to support causal inferences. Sampling rigor and image-analysis details need to be fully specified

Comments to the Author:

- The study addresses a clinically relevant problem with a solid human design and clear age-related differences in inflammation and collagen remodeling during reloading. The central message is promising, but the manuscript requires substantial methodological tightening, clearer phenotype definitions, fuller statistical transparency, and additional analyses to align with the standards of The Journal of Physiology.

- The macrophage phenotyping needs greater specificity or more cautious terminology. As Referee #1 notes, CD68 is pan-myeloid, and CD206 alone does not define an alternatively activated phenotype. Either expand the panel to include at least one additional marker for each state, such as HLA-DR or CD86 for classically activated cells and CD163 for alternatively activated cells, or relabel all instances to neutral descriptors ("CD68⁺CD206⁻ macrophages" and "CD68⁺CD206⁺ macrophages") and temper any mechanistic language linked to "M1/M2". Similarly, in tissue senescence analyses, you identify "senescent macrophages" by CD11b with p21. CD11b is not macrophage-specific in human tissue; either add CD68 to the co-stain or consistently relabel this

compartment as "senescent myeloid cells," including in the abstract and conclusions. This point aligns with Referee #2's concern regarding Fig. 5.

We recognize that CD206 alone provides limited information regarding macrophage alternative activated phenotype. To address this, we removed M1/pro-inflammatory and M2/anti-inflammatory terminology and instead used CD68⁺ CD206⁻ and CD68⁺ CD206⁺ descriptors. Based on established literature in human literature, we maintain that CD68⁺ cells represent human macrophages, as supported by Saclier *et al.* In these experiments, the authors conducted detailed cell culture experiments to identify macrophages with CD68 and demonstrated CD68 expression in skeletal muscle macrophages (Saclier *et al.*, 2013; Saclier *et al.*, 2017).

We also recognize that CD11b is a pan-myeloid marker and not elusively specific to macrophages. However, in human skeletal muscle, we believe that the majority of the CD11b⁺ cells are macrophages. In fact, Kosmac *et al.* found that CD68 and CD11b are comparable as a pan-macrophage marker in human skeletal muscle (Kosmac *et al.*, 2018). Therefore, we feel justified to maintain the language of using CD11b for labeling human macrophages in this study.

- The time-course of "total infiltration" versus lineage-defined counts must be reconciled. Referee #2 points out that the H&E-based infiltration curve does not map cleanly to macrophage accrual in young adults and that neutrophil responses appear surprisingly small. Validate neutrophil detection with a human-appropriate marker such as CD66b or neutrophil elastase in addition to MPO, and confirm that staining quality and quantification are robust. It seems adding a pan-leukocyte index (for example, CD45⁺ cells per area) across all time points to harmonize H&E "infiltration" with immune-lineage counts would be helpful, and clarify which non-macrophage populations account for the early post-reloading signal in young adults if macrophages do not.

That is an important observation and merits discussion. We believe that some of the discrepancy between Fig 2D and 3A can be attributed to differences in sample size analyzed for the H&E assay. Initially, we examined only a subset of participants from each group (Fig 2D; YA: n=10, OA: n=10) in the H&E assay, whereas the macrophage dataset included all participants (Fig 3A; YA: n=18, OA: n=18). Consequently, direct comparison between these figures may appear inconsistent. To address this, we analyzed the remaining participants for the H&E assay, and the results now align more closely with the macrophage data. Additionally, we performed Pearson's correlation (controlled for FDR) and the H&E analyses were significantly associated with CD68 and MPO during the recovery. See data below regarding these correlations. These data have been added to the manuscript.

Baseline

H&E vs CD68: $r=0.32$, $P=0.060$

H&E vs MPO: $r=0.23$, $P=0.183$

Post-Immobilization

H&E vs CD68: **$r=0.43$, $P=0.010$**

H&E vs MPO: $r=0.18$, $P=0.311$

2d-recovery

H&E vs CD68: **$r=0.60$, $P<0.001$**

H&E vs MPO: **$r=0.38$, $P=0.029$**

7d-recovery

H&E vs CD68: **$r=0.67$, $P<0.001$**

H&E vs MPO: **$r=0.49$, $P=0.003$**

We also conducted a CD45 immunofluorescence assay with a subset of participants (YA: $n=10$ (5M/5F), OA: $n=10$ (5M/5F)) and validated that the H&E mainly captures immune cells (Fig S2). Below are the correlations between the data from the H&E and CD45 at each time point which also have been added to the manuscript.

Baseline

H&E vs CD45: $r=0.29$, $P=0.237$

Post-Immobilization

H&E vs CD45: **$r=0.62$, $P=0.005$**

2d-recovery

H&E vs CD45: $r=0.37$, $P=0.118$

7d-recovery

H&E vs CD45: **$r=0.68$, $P=0.001$**

Regarding the modest neutrophil response during recovery following immobilization, this likely reflects the nature of the experimental model. Neutrophil infiltration typically occurs in response to substantial muscle damage or injury. Although we observed minor

signs of muscle damage and regeneration, and particularly in older adults, muscle reloading following limb immobilization does not represent a traditional model of muscle damage mode. Therefore, this model is less likely to elicit a robust neutrophil response during recovery. Additionally, after muscle damage, the neutrophil infiltration, action, and clearance is normally quite rapid which may have been missed the current sampling time points. Nonetheless, we have conducted another neutrophil assay (CD66b, Fig S3) on a subset of participants (YA: n=10 (5M/5F); OA: n=10 (5M/5F) and similarly confirm a modest neutrophil response to recovery.

- The age bias in late inflammation requires either additional data or tighter interpretation. Please provide evidence for increased recruitment at day 7 in older adults. If tissue permits, include a recruitment-associated marker such as CCR2 on CD68⁺ cells at day 7; if not feasible, state this limitation and frame late recruitment as a hypothesis. If you propose impaired immunological tolerance, provide an exploratory FOXP3⁺ Treg count in the tissue or clearly flag this as speculation and add appropriate citations. The discussion should articulate a mechanistic model that is consistent with your human data while acknowledging where evidence is currently associative.

We do not interpret the age-differences in immune cell infiltration to be a results of “late” recruitment but rather a more robust infiltration of immune cells – particularly CD68⁺ CD206⁺ macrophages in older adults. We have removed any language that suggest otherwise. The young cohort served as a control did not exhibit an “earlier” immune cell infiltration which rules out possible age differences in the recruitment timecourse, at least as determined from our day 2 recovery time point. Rather young subjects during recovery exhibited a mild and appropriate immune infiltration response, whereas the older adults demonstrated a more exaggerated infiltration response at this same time point. However, we added to the discussion speculating that the older adults may exhibited an increased and possibly dysregulated recruitment of macrophages during the muscle recovery (e.g., CCR2). Unfortunately, we cannot confirm this hypothesis due to lack of muscle tissue sample but warrants further investigation.

- The collagenolysis experiment needs controls, method clarity, and reframed claims. Please define all assay controls explicitly, explain how negative values arise in the DQ-collagen readout, and report background subtraction procedures. Validate monocyte-to-macrophage differentiation and polarization in human cells with simple phenotypic checks (for example, CD14/CD68 upshift; MRC1/CD206, HLA-DR or TNF expression) and report differentiation efficiency by donor age. The donor selection strategy for this assay is biased toward “high responders”? If so, it is necessary to state that it is

exploratory and show all individual points, report effect sizes with confidence intervals, and soften causal language. If possible, add p21 or SA- β -Gal to the cultured macrophages to support the proposed link between myeloid senescence and reduced collagen degradation in older adults. These adjustments address Referee #2's Major Comment on Fig. 6.

Thank you for your feedback on this data set. We acknowledge that these data were biased and limited; therefore, we determined they were too preliminary, and we have removed the *in vitro* data previously presented in Fig. 6.

- Statistical reporting and outlier handling must be explicit. Confirm that assumptions for ANOVA, t-tests, and Pearson correlations were tested for all variables, state the diagnostics used, and describe remedies where violated, including any transformations or non-parametric alternatives. Specify the Grubbs' test parameters used for outlier removal, including alpha and whether one- or two-sided, and state whether the rule was pre-specified before viewing the data. Where you present many correlations, indicate whether you controlled the false discovery rate and, if not, provide a rationale. In tables, please denote significant main effects as well as interactions, as suggested by Referee #2.

In the *Statistical Analyses* subsection, we added details to improve transparency regarding our methods. Specifically, we confirmed assumptions for the 2-way ANOVAs/mixed effects analyses, as well as the Pearson's correlations. Additionally, we provided parameters for the Grubbs' test used to identify and remove outliers. We also denoted the significant main effects as well as interactions on the figures and tables.

- Sex as a biological variable needs to be handled transparently. Please acknowledge established sex differences in innate and adaptive immunity, and then discuss how these might influence your outcomes. Whether the menstrual cycle phase was controlled in young female participants; if not, state this plainly as a limitation and explain the potential impact on inflammatory readouts.

Thank you for your feedback regarding sex as a biological variable. We have now included sex-specific comparison for all analyses in the supplemental material. We have also added a limitation paragraph to the Discussion (lines 383-393) as we did not monitor or control for menstrual cycles. We also added discussion how the inflammatory profile of women changes after menopause and the effects of estrogen on the inflammatory environment and how this may partly contribute to the observed outcomes.

- Sampling rigor and image analysis need more detail. Report the number of sections or areas quantified per biopsy for each stain, the region-of-interest selection rules, the analyzed area per section, and any exclusion criteria for fields. Please make sure all figure axes and legends include units and normalization. For the large variance in older adults at day 2 in Fig. S1A-B, please follow Referee #2's suggestion: stratify participants into high versus low dystrophin-negative responders at day 2 and test whether this stratification predicts day-7 macrophage accrual, myeloid-cell senescence, and collagen content. Report the results regardless of outcome and comment on whether this explains variance in later endpoints.

In the methodology, we have now added the subsection *Image Analyses* to describe how images were evaluated. We also verified that all figure axes and legends include appropriate units and normalization details.

We completed Referee #2's suggestion by stratifying participants into high vs low dystrophin-negative responders at day 2 and performed Pearson's correlations with macrophage content, senescent macrophages, and collagen content at day 7. Unfortunately, these analyses did not unmask relationships between these variables within either the high- or low-responder groups. Please see the data below.

High Responders

Dystrophin⁻ vs CD68⁺: $r=-0.19$, $P=0.680$

Dystrophin⁻ vs CD68⁺ CD206⁻: $r=0.22$, $P=0.633$

Dystrophin⁻ vs CD68⁺ CD206⁺: $r=-0.50$, $P=0.248$

Dystrophin⁻ vs CD11b⁺ p21⁺: $r=-0.33$, $P=0.517$

Dystrophin⁻ vs Sirius Red: $r=0.55$, $P=0.201$

Low Responders

Dystrophin⁻ vs CD68⁺: $r=0.10$, $P=0.748$

Dystrophin⁻ vs CD68⁺ CD206⁻: $r=0.17$, $P=0.574$

Dystrophin⁻ vs CD68⁺ CD206⁺: $r=0.04$, $P=0.909$

Dystrophin⁻ vs CD11b⁺ p21⁺: $r=0.24$, $P=0.530$

Dystrophin⁻ vs Sirius Red: $r=-0.01$, $P=0.962$

Even though a few of the older adults exhibited signs of muscle damage (Dystrophin⁻ fibers), as mentioned above, muscle reloading is not a robust model of muscle damage and thus very few dystrophin negative fibers during muscle recovery were observed. This outcome is also consistent with the limited changes in muscle damage detected by neutrophil markers.

- The PBMC culture protocol needs to state cell composition and purity. Clarify whether lymphocytes were removed before or after seeding. If not removed, explain how the culture conditions favored monocyte-derived macrophages (for example, M-CSF reliance and adherence selection) and provide a brief purity estimate or marker check.

As mentioned above we have removed the PBMC cell culture experiment data from Figure 6 due to the data being too premature.

- Textual accuracy and framing should be corrected throughout. Where the manuscript currently states that age differences emerge "in the early days following immobilization," revise to reflect that the clearest divergence occurs late in the reloading week, consistent with your data. Replace any causal phrasing that implies disruption or impairment by a particular cell phenotype with association language unless supported by direct manipulation. Ensure that each figure legend lists the exact n per group at each time point and that normalization conventions are consistent across panels.

We refined our language when making inferences at 7d-recovery by removing the term "early" and eliminating phrases that implied cellular dysfunction or impairment. Additionally, we added sample sizes for each group in the figure legends and ensured that normalization methods are consistent across panels.

- Because several points hinge on marker specificity and cell identity in human tissue, I expect either expanded panels or neutral relabeling to avoid over-interpretation. Because several claims rely on correlations, I expect fuller statistical transparency, clear acknowledgment of multiple testing where relevant, and effect sizes with uncertainty, not only P-values. Because you draw mechanistic inferences from an underpowered and selectively sampled ex vivo assay, I expect the claims to be reframed as exploratory/association unless you can add the requested controls and validation.

We appreciate the feedback, which we believe has strengthened the manuscript. We incorporated changes consistent with established literature in human skeletal muscle immunology. Specifically, we replaced the M1/pro-inflammatory and M2/anti-inflammatory terminology with CD68⁺ CD206⁻ and CD68⁺ CD206⁺ descriptors. Based on

prior reports, we maintain that the pan-myeloid markers CD68 and CD11b predominantly identify macrophages in human muscle.

Additionally, we improved transparency by providing details on our statistical models and tests. Finally, we removed our *in vitro* PBMC dataset, as we determined these data were too preliminary.

- Please make sure the Standard deviation (SD) is reported, not SEM.

For all the figures we replaced SEM with SD and included the individual data points on the delta from baseline figures.

Referee #1:

The manuscript compares the dynamics of myogenic and non-myogenic cells (macrophages, FAPs), cellular senescence, and muscle collagen content during early recovery (2 and 7 days) following 14 days of unilateral limb immobilization in young (n = 18; ~24 years) and older (n = 18; ~69 years) male and female adults. Overall, the study is well designed and clearly presented, and the topic is timely. Detailed comments are provided below.

1 - Why was no additional marker used to determine the pro-inflammatory phenotype of macrophages? For example, it is possible that within the CD68+ population there are macrophages expressing other "anti-inflammatory" markers and lacking CD206 expression (e.g., CD68⁺CD163⁺). This phenotypic complexity can be crucial for the interpretation of the results.

Thank you for this thoughtful comment. We agree that macrophage phenotypes are complex and that additional markers such as CD163 could provide further resolution. However, due to technology limitations, we were limited to four markers per section – two for muscle fiber visualization and DAPI for nuclear staining – leaving only two markers available for macrophage identification and phenotyping. Within these constraints, we selected CD68 and CD206 based on their widespread use in human skeletal muscle immunology and their ability to distinguish macrophage subpopulation. This approach is consistent with methodologies employed by other human skeletal muscle experts in the field (Kosmac *et al.*, 2018; Sorensen *et al.*, 2019; Ahmadi *et al.*, 2022). Additionally, we have a limited amount of muscle tissue samples. We prioritized performing CD206 and therefore we could not conduct another anti-inflammatory phenotyping procedure. To minimize categorizing macrophages into classical and anti-inflammatory populations, we have resorted to using terminologies of CD68⁺ CD206⁻ and CD68⁺ CD206⁺ descriptors instead.

2 - In the PBMC culture protocol, it is not clear whether the lymphocytes were removed from the wells. Could you please clarify this point?

After further consideration, we determined that this dataset was too preliminary and we removed the *in vitro* PBMC data from the revised manuscript.

3 - Were the statistical assumptions for the analysis of variance, t-test, and Pearson's correlation met for all variables? If so, it is important to specify this in the manuscript.

Yes, all statistical assumptions for the ANOVA and Pearson's correlations were met. To improve transparency, we added sentences in the *statistical analyses* specifying that these assumptions were verified and included additional details on the statistical models used.

4 - There are well-established differences in both innate and adaptive immune functions, with women generally exhibiting stronger immune responses but also a greater susceptibility to autoimmune and inflammatory diseases. In this context, it would be important to consider whether sex-based differences exist in the measurements of both immune and non-immune variables examined in the study. Given the relevance of this issue, I suggest including a dedicated section on limitations to address this point. Additionally, providing the main measurements disaggregated by sex as supplementary material would substantially strengthen the manuscript.

Thank you for this important suggestion. We have added a paragraph in the Discussion (lines 383-393) acknowledging sex as a biological variable and noting the limitation that we did not monitor or control for menstrual cycles or age-related changes in inflammatory conditions, which may have influenced our results. We also discuss how inflammatory profiles change after menopause and the role of estrogen in modulating the inflammatory environment and how this may have shaped the existing data. Furthermore, we included sex-specific comparisons for all measurements in the supplemental material.

5 - Still regarding sex differences, was the menstrual cycle phase considered during the recruitment of the young female participants? If not, this limitation should be clearly mentioned in the manuscript.

No, menstrual cycle was not controlled or documented, and we added it as a limitation within the discussion as mentioned above.

Referee #2:

This clinical trial conducted by Skiles and colleagues was designed to compare cellular dynamics (of satellite cells, FAPs, and various subtypes of immunocytes), cellular senescence, and collagen content in human limb muscle (vastus lateralis) over a timecourse of unilateral immobilization followed by reloading. The authors uncovered excessive inflammatory responses to reloading in older adults (OA) compared to young adults (YA). In particular, macrophage accumulation was exaggerated in OA late after reloading (7d); such accumulation was positively correlated with senescence in the muscle myeloid-cell compartment as well as collagen content - both of which were amplified by reloading. These findings are consistent with the comparatively larger body of literature from preclinical models of the effects of aging on cellular responses to unloading/loading and acute injury. As older populations are commonly hospitalized due to illness or injury (and the frequency of falls is on the rise), clinical trials on disuse-associated muscle atrophy, inflammation, and fibrosis are of high importance. While the correlations presented in this work highlight avenues for future mechanistic studies, there are several concerns that should be addressed to strengthen the authors' claims and provide greater clarity.

Major comments

1) The shapes of the response curves in Fig. 2D (total infiltration), Fig. S2A (neutrophils) and Fig. 3A (macrophages) are quite different, particularly for YA. Do the authors have any insights into how to harmonize these? A few suggestions for possible explanations:

- Macrophages are the predominant cell type in muscle, so one would expect that the curves in 3A would map nicely to 2D; however, this doesn't appear to be true, at least for YA. Do the authors have any data or insights into what other cell types could account for the greater infiltration in POST and 2d not accounted for by the macrophage compartment?

Thank you for this insightful observation. We agree that the differences in curve shapes warrant discussion. We believe that part of the discrepancy between Fig 2D and 3A can be contributed to differences in sample size analyzed for the H&E assay. Initially, we examined only a subset of participants (Fig 2D; YA: n=10, OA: n=10), whereas the macrophage dataset included all participants (Fig 3A; YA: n=18, OA: n=18). To address this, we analyzed the remaining participants for the H&E assay, and the results align more closely with the macrophage data. Additionally, we performed Pearson's

correlation (controlled for FDR) and the H&E analyses were significantly associated with CD68 and MPO labeled cells during the recovery. See data below regarding these correlations.

BX1

H&E vs CD68: $r=0.32$, $P=0.060$

H&E vs MPO: $r=0.23$, $P=0.183$

BX2

H&E vs CD68: **$r=0.43$, $P=0.010$**

H&E vs MPO: $r=0.18$, $P=0.311$

BX3

H&E vs CD68: **$r=0.60$, $P<0.001$**

H&E vs MPO: **$r=0.38$, $P=0.029$**

BX4

H&E vs CD68: **$r=0.67$, $P<0.001$**

H&D vs MPO: **$r=0.49$, $P=0.003$**

We also conducted a CD45 immunofluorescence analysis in a subset of participants in each group (Fig S2; YA: $n=10$ (5M/5F), OA: $n=10$ (5M/5F) and, similarly, this label corresponded to the H&E stain suggesting that the infiltrating cells are indeed immune cells in the tissue. Below are the correlation between H&E and CD45 at each time point.

Baseline

H&E vs CD45: $r=0.29$, $P=0.237$

Post-Immobilization

H&E vs CD45: **$r=0.62$, $P=0.005$**

2d-recovery

H&E vs CD45: $r=0.37$, $P=0.118$

7d-recovery

H&E vs CD45: **$r=0.68$, $P=0.001$**

- Related to the point above, the neutrophil responses are surprisingly minor, particularly in the early recovery window. One would expect that perhaps neutrophils could harmonize the shapes of the curves in 2D and 3A, but that does not appear to be true in this case. Is it possible that the MPO staining did not work properly (the area/size and shape of staining do not look as expected)? The authors should confirm with another marker such as Ly6G.

Regarding the neutrophil response, we believe the minimal infiltration is multifaceted. Muscle reloading following immobilization does not represent a traditional muscle damage model, which typically elicits robust neutrophil infiltration (e.g., electrically stimulated muscle contractions, eccentric exercise, or injury). Although older adults exhibit minor signs of muscle damage/regeneration, it is variable, and the overall lack of substantial damage likely explains the limited neutrophil response. We also conducted CD66b immunofluorescence to further validate the minor neutrophil response during recovery in a subset of participants in each group (Fig S3; YA: n=10 (5M/5F), OA: n=10 (5M/5F)). Together, we believe that macrophages make up a large fraction of these immune cells that infiltrate into older adult skeletal muscle during recovery.

2) The protracted inflammatory responses in muscles from OA vs YA is striking (i.e., differences on day 7 of recovery). What do the authors think is the basis of this observation? This point warrants discussion in the text and perhaps additional assays:

- Based on data from preclinical models, it's likely that there is more recruitment on d7 in OA vs YA. Do the authors have any evidence that this is true in their study? There are a few ways to address this, but one would be to add CCR2 to the panel in Fig. 3.

We agree that the robust inflammatory response in older adults at day 7 is noteworthy and it is possible that the older adults had elevated signaling for macrophage recruitment leading to the robust macrophage response during the recovery. There is evidence that inhibiting recruitment such as CCR2 can improve muscle recovery (Blanc *et al.*, 2020). Adding CCR2 or other recruitment markers would provide valuable insight; however, due to limited amount muscle tissue samples we are not able to assess recruitment. We have added to the discussion the possibility of age-related recruitment changes perhaps regulating this immune infiltration response.

- Active recruitment late in recovery in OA could be due to defects in mechanisms of peripheral immunological tolerance (i.e., Tregs). Do the authors have any evidence for this?

During muscle repair from injury, regulatory T (T_{reg}) cells play a critical role by limiting excessive buildup of inflammatory myeloid cells and preventing fibrosis in regenerative muscle (Schiaffino *et al.*, 2017; Panduro *et al.*, 2018). T_{reg} cells also promote the transition of macrophages from pro-inflammatory to anti-inflammatory phenotypes and support the proliferation and activation of satellite cells (Panduro *et al.*, 2018). In aged muscle, T_{reg} cell accumulation is markedly reduced due to impaired recruitment, proliferation, and retention which likely contributes to the slowed or incomplete regenerative response often observed with aging (Schiaffino *et al.*, 2017). However, muscle reloading from disuse atrophy is not a traditional muscle damage/regenerative model and we do not suspect that T_{reg} cells have an influential role in the muscle recovery from disuse.

3) There are a few important issues in Figure 5:

- CD11b alone is not sufficient to identify macrophages, as it is highly expressed by other myeloid populations too (e.g., neutrophils). The authors should either enhance their panel or change "macrophages" to "myeloid cells" when referencing these data.

Thank you for this comment. In human skeletal muscle, the majority of the CD11b⁺ cells are macrophages. In fact, Kosmac *et al.* found that CD68 and CD11b are comparable as a pan-macrophage marker in human skeletal muscle (Kosmac *et al.*, 2018). Therefore, we feel justified to maintain the language of using CD11b to classify pan macrophages in this study.

- The correlations in Figure 5G,H are less convincing than co-staining with the panel used in Fig. 3.

We have removed the correlation between senescent macrophages (CD11b⁺ p21⁺ DAPI⁺) and CD68⁺ CD206⁻ and CD68⁺ CD206⁺ macrophages at the 7d-recovery and replaced with a correlation between infiltrating macrophages (CD68⁺ DAPI⁺) and cellular senescence (SA- β -Gal⁺) at the 7d-recovery. This data suggest that the amount of infiltrating macrophages is associated with the amount of cellular senescence during the muscle recovery.

4) The results in Fig. 6G-J are interesting and quite surprising. A few comments:

- The reviewer could not find the controls for this assay defined anywhere. Please add a definition in the methods and/or figure legend and clarify how one derives negative collagen degradation values in this fluorescent collagen assay.

- The surprise from this data comes from 1) the very small difference in collagen degradation between each macrophage state within an age group and 2) the lower degradation in OA independent of age. Please provide validation of the differentiation efficiency for each age. Also, it would strengthen the conclusion of the paper and help explain surprise #2 if the authors could show that macrophages derived from OA monocytes are more senescent in their system.

Thank you for the feedback on this figure and after further review of the comments from the editor and the other reviewers we decided that the *in vitro* data is too preliminary and was removed from the manuscript.

5) There is large variance within the OA day 2 group in Fig. S1A,B. Can individuals be binned as high or low responders based on dystrophin-neg. fibers at this time? If so, does the fraction of dystrophin-neg. fibers in the OA day 2 condition correlate with anything on day 7, particularly accumulation of macrophages, senescent cells, and collagen?

Thank you for the suggestion. We separated the high responders from the low responders from the dystrophin negative fiber data set (2d-recovery) and performed Pearson's correlations with the macrophage content, cellular senescence, and collagen content (7d-recovery) and we did not find any significant associations. Please see the data below regarding these correlations.

High Responders

Dystrophin⁻ vs CD68⁺: $r=-0.19$, $P=0.680$

Dystrophin⁻ vs CD68⁺ CD206⁻: $r=0.22$, $P=0.633$

Dystrophin⁻ vs CD68⁺ CD206⁺: $r=-0.50$, $P=0.248$

Dystrophin⁻ vs CD11b⁺ p21⁺: $r=-0.33$, $P=0.517$

Dystrophin⁻ vs Sirius Red: $r=0.55$, $P=0.201$

Low Responders

Dystrophin⁻ vs CD68⁺: $r=0.10$, $P=0.748$

Dystrophin⁻ vs CD68⁺ CD206⁻: $r=0.17$, $P=0.574$

Dystrophin⁻ vs CD68⁺ CD206⁺: $r=0.04$, $P=0.909$

Dystrophin⁻ vs CD11b⁺ p21⁺: $r=0.24$, $P=0.530$

Dystrophin⁻ vs Sirius Red: $r=-0.01$, $P=0.962$

Additional comments

1) Please specify threshold used for Grubb's test.

In the methods under *Statistical Analyses* we added details to our statistical models including the threshold for Grubb's test.

2) It would be helpful to denote significant main effects (perhaps using letters) in the tables, e.g. main effect of age on MHCIIA CSA (lines 261-262).

We have now added significant main effects to the figures and tables.

3) The authors seem to have mistakenly said "in the early days following immobilization" on line 278 instead of "in the late phase of recovery" or something similar. The data in Fig. 2D clearly indicate a protracted inflammatory response in OA vs YA, not a difference in early accrual.

We apologize for this confusion. We have now removed the "early" terminology when describing changes that occur at the 7d-recovery.

4) How many sections or areas were analyzed per biopsy per stain/analysis?

In the methodology, we added the subsection *Image Analyses* where we provided details on how we analyzed the images for each of the assays we performed.

5) Fig. S1A,B lacks a legend for YA and OA groups

Thank you for pointing that out. We added a legend to Fig S1.

REFERENCES

- Ahmadi M, Karlsen A, Mehling J, Soendenbroe C, Mackey AL & Hyldahl RD. (2022). Aging is associated with an altered macrophage response during human skeletal muscle regeneration. *Exp Gerontol* **169**, 111974.
- Blanc RS, Kallenbach JG, Bachman JF, Mitchell A, Paris ND & Chakkalakal JV. (2020). Inhibition of inflammatory CCR2 signaling promotes aged muscle regeneration and strength recovery after injury. *Nat Commun* **11**, 4167.
- Kosmac K, Peck BD, Walton RG, Mula J, Kern PA, Bamman MM, Dennis RA, Jacobs CA, Lattermann C, Johnson DL & Peterson CA. (2018). Immunohistochemical Identification of Human Skeletal Muscle Macrophages. *Bio Protoc* **8**.
- Panduro M, Benoist C & Mathis D. (2018). T(reg) cells limit IFN-gamma production to control macrophage accrual and phenotype during skeletal muscle regeneration. *Proc Natl Acad Sci U S A* **115**, E2585-E2593.
- Saclier M, Theret M, Mounier R & Chazaud B. (2017). *Effects of Macrophage Conditioned-Medium on Murine and Human Muscle Cells: Analysis of Proliferation, Differentiation, and Fusion*, vol. 1556. Springer Science+Business Media LLC.
- Saclier M, Yacoub-Youssef H, Mackey AL, Arnold L, Ardjoune H, Magnan M, Sailhan F, Chelly J, Pavlath GK, Mounier R, Kjaer M & Chazaud B. (2013). Differentially activated macrophages orchestrate myogenic precursor cell fate during human skeletal muscle regeneration. *Stem Cells* **31**, 384-396.
- Schiaffino S, Pereira MG, Ciciliot S & Rovere-Querini P. (2017). Regulatory T cells and skeletal muscle regeneration. *FEBS J* **284**, 517-524.
- Sorensen JR, Kaluhiokalani JP, Hafen PS, Deyhle MR, Parcell AC & Hyldahl RD. (2019). An altered response in macrophage phenotype following damage in aged human skeletal muscle: implications for skeletal muscle repair. *FASEB J* **33**, 10353-10368.

Dear Dr Drummond,

Re: JP-RP-2026-290346R1 "**Robust skeletal muscle immune cell infiltration and macrophage senescence occur within a week of recovery after limb immobilization in older adults**" by Chad M Skiles, Zachary James Fennel, Paul-Emile Bourrant, Elena M Yee, Robert J Castro, Hannah A Zabriskie, Melina Itinose, Mark A Supiano, Ryan O'Connell, and Micah J Drummond

We are pleased to tell you that your paper has been accepted for publication in The Journal of Physiology.

Yours sincerely,

Karyn Hamilton
Senior Editor
The Journal of Physiology

IMPORTANT POINTS TO NOTE FOLLOWING ACCEPTANCE OF YOUR PAPER:

- **IMPORTANT NOTICE ABOUT OPEN ACCESS:** To assist authors whose funding agencies mandate immediate public access to published research findings, The Journal of Physiology allows authors to pay an Open Access (OA) fee to have their papers made freely available immediately on publication.

The Corresponding Author will receive an email from Wiley with details on how to register or log in to Wiley Authors where you will be able to place an order.

- You can check if your funder or institution has a Wiley Open Access Account here:
<https://authors.wiley.com/author-resources/Journal-Authors/open-access/author-compliance-tool.html>

- You can help your research get the attention it deserves! Check out Wiley's free Promotion Guide for best-practice recommendations for promoting your work at: www.wileyauthors.com/eoo/guide. You can learn more about Wiley Editing Services which offers professional video, design, and writing services to create shareable video abstracts, infographics, conference posters, lay summaries, and research news stories for your research at: www.wileyauthors.com/eoo/promotion.

- If you would like to receive our 'Research Roundup', a monthly newsletter highlighting the cutting-edge research published in The Physiological Society's family of journals (The Journal of Physiology, Experimental Physiology, Physiological Reports, The Journal of Nutritional Physiology and The Journal of Precision Medicine: Health and Disease), please click this link, fill in your name and email address and select 'Research Roundup':
<https://www.physoc.org/journals-and-media/membernews>

EDITOR COMMENTS

Reviewing Editor:

Comments for Authors to ensure the paper complies with the Statistics Policy (Required):

Please ensure that all figures include individual data points for $n < 30$. Please provide precise p-values in the figures, and harmonize the fonts/sizes across all panels (e.g., Figure 6D and 6E).

Comments to the Author (Required):

Thank you for responding to the comments in a thorough and constructive manner.

Nevertheless, please ensure that the revised manuscript fully complies with the journal's guidelines regarding data presentation and statistical reporting, particularly with respect to the display of individual data points and the reporting of exact p-values.

Senior Editor:

Thank you for submitting your revised Journal Club discussion. We are pleased to accept it for publication in The Journal of Physiology. I noticed that you refer to your tables as "Table S1, S2", etc. The Journal doesn't typically accept supplementary content (see Information For Authors) so the staff may be getting in touch to have you change these to Table 1, Table 2, etc. Otherwise, we appreciate your revisions and thank you for your interest in The Journal of Physiology. Congratulations!

[NOTE FROM EDITORIAL OFFICE: we will move your 'supplemental tables' into an Appendix at the end of the article, and rename them Table A1, A2, etc. - please check everything carefully at proof stage.]

REFeree COMMENTS

Referee #1:

The authors have either revised the manuscript or provided satisfactory explanations addressing the points raised by the reviewers. The implemented changes have strengthened the manuscript, making it suitable for publication in The Journal of Physiology.

Referee #2:

The authors have adequately addressed all comments, and their manuscript is considerably enhanced.